# Multi-wavelength dataset of aerosol extinction profiles retrieved from GOMOS stellar occultation measurements

Viktoria F. Sofieva[1], Monika Szelag[1], Johanna Tamminen[1], Didier Fussen[2], Christine Bingen[2], Filip Vanhellemont[2], Nina Mateshvili[2], Alexei Rozanov[3] and Christine Pohl[3]

[1] Finnish Meteorological Institute, Helsinki, Finland
[2] Royal Belgian Institute for Space Aeronomy (BIRA-IASB), Brussels, Belgium
[3] Institute of Environmental Physics, University of Bremen, Bremen, Germany

*Correspondence to*: Viktoria Sofieva (viktoria.sofieva@fmi.fi)

**Abstract.**

In this paper, we present the new multi-wavelength dataset of aerosol extinction profiles, which are retrieved from the averaged transmittance spectra by the Global Ozone Monitoring by Occultation of Stars instrument on board the Envisat satellite.

Using monthly and zonally averaged transmittances as a starting point for the retrievals enables us to improve the signal-to-noise ratio and eliminate possible modulation of transmittance spectra by uncorrected scintillations. The two-step retrieval method is used: the spectral inversion is followed by the vertical inversion. The spectral inversion relies on the removal of contributions from ozone, $NO_2$, $NO_3$ and Rayleigh scattering from the optical depth spectra, for each ray perigee altitude. In the vertical inversion, the profiles of aerosol extinction coefficients at several wavelengths are retrieved from the collection of slant aerosol optical depth profiles.

The retrieved aerosol extinction profiles (FMI-GOMOSaero dataset v1) are provided in the altitude range 10-40 km at wavelengths 400, 440, 452, 470, 500, 525, 550, 672 and 750 nm, for the whole GOMOS operating period from August 2002 to March 2012.

Extensive intercomparisons of the retrieved FMI-GOMOSaero aerosol profiles with aerosol profile data from other satellite instruments at several wavelengths have been performed. It is found that the average difference between FMI-GOMOSaero and other datasets is within 20-40% in the lower and middle stratosphere, the standard deviation ~ 20-50%, and the correlation coefficient of time series 0.65-0.85.

The created FMI-GOMOSaero dataset can be used in merged datasets of stratospheric aerosols. It might be also used as a priori information for satellite retrievals in 2002-2012.

# 1    Introduction

Stratospheric aerosols impact the radiative forcing and thus the energy balance of the Earth's atmosphere, therefore information about their distribution and variability is of high importance for climate related studies. The stellar occultation instrument GOMOS (Global Ozone Monitoring by Occultation of Stars) , which operated on the Envisat satellite in 2002-2012, was capable of providing the profiles of aerosol extinction coefficient  in the altitude range from ~10 to 40 km (Kyrölä et al., 2010; Bertaux et al., 2010; Vanhellemont et al., 2010). The GOMOS aerosol data have already been used in the
merged stratospheric aerosol dataset (Vernier et al., 2011) and climate studies (Solomon et al., 2011; Santer et al., 2014; Brühl et al., 2018; Schallock et al., 2021), and to produce reference climate data records (Popp et al., 2016; Bingen et al., 2017).

GOMOS was equipped with UV-VIS (250 - 690 nm) and IR (755-774 nm and 926-954 nm) grating spectrometers operating at a sampling frequency of 2 Hz, and with two fast photometers at ~472 nm and ~672 nm operating at 1 kHz sampling
frequency. GOMOS measured stellar irradiance in the limb-viewing geometry from ~140 km down to full extinction at about 10-15 km (Bertaux et al., 2010). The transmittance spectra are obtained after dividing the stellar spectra observed through the atmosphere by the reference spectrum above the atmosphere. After correction of refraction effects (refractive attenuation and scintillation), the transmittance spectra are the basis for retrievals of trace gases and aerosol extinction.

The GOMOS IPF (Instrument Processor Facility) v6 retrievals of ozone, $NO_2$, $NO_3$ and aerosol extinction rely on two-step
inversion (Kyrölä et al., 2010). First, the horizontal column densities are retrieved from transmittance spectra $T(\lambda,z)$ for each tangent altitude (the spectral inversion).  After that, the local density profiles are obtained from the profiles of horizontal column densities (the vertical inversion). In the IPF spectral inversion, the contribution due to Rayleigh scattering is estimated using the ECMWF data, and horizontal column densities of ozone, $NO_2$, $NO_3$ and aerosol extinction are retrieved simultaneously from UV-VIS spectrometer data, with iterative DOAS-type inversion for $NO_2$ and $NO_3$. Since the spectral
dependence of aerosol cross-sections, which  depend on particle size distribution, is not known, the IPF v6 processor uses  a parameterization via a second-degree polynomial model in wavelength $\lambda$ (Kyrölä et al., 2010; Vanhellemont et al., 2010). The GOMOS IPF v6 processor provides the aerosol extinction coefficient data record at 500 nm and its spectral dependence, expressed by the coefficients of the polynomial mentioned above.

In the AERGOM processor, which is optimized for aerosol retrievals (Vanhellemont et al., 2016), the data from the
spectrometer B1 (755-774 nm) are also used together with the UV-VIS spectrometer data for retrievals of aerosol extinction. In AERGOM, the spectral dependence of aerosol extinction is parameterized by a second-degree polynomial in $1/\lambda$. As in IPF v6, the retrievals are performed in two steps, the non-linear spectral inversion is followed by the linear vertical inversion. In the AERGOM spectral inversion, horizontal column densities of ozone, $NO_2$, $NO_3$ and aerosol extinction are retrieved simultaneously from UV-VIS spectrometer data. Covariances between species after the spectral inversion are taken into
account in AERGOM, while they are neglected in the IPF v6 processor. AERGOM performs the vertical inversion for all species simultaneously, while the IPF v6 processor does it for each species separately, with a decreased information content.

Finally, for aerosol extinction, AERGOM applies three altitudinal regularization constraints (at three wavelengths), while IPF v6 applies only one (aerosol extinction at 500 nm). From all AERGOM vertical aerosol extinction coefficient profiles, a 5-days gridded climate data record with a spatial resolution of 5° latitude and 60° longitude was created (Bingen et al.,

2017). This climate data record, which is referred to as CCI-AerGOM and is available at https://cds.climate.copernicus.eu/cdsapp#!/dataset/10.24381/cds.239d815c?tab=form, contains the profiles of average aerosol extinction coefficient at 355, 440, 470, 550 and 750 nm.

Although GOMOS aerosol extinction coefficient data are available from the ESA IPF v6 and AERGOM processors, the reported spectral dependence of the aerosol extinction is often not realistic, mainly due to interference with ozone retrievals,

limited wavelength range in the IPF v6 retrievals, insufficient signal-to-noise ratio, residual scintillation and the limitations of the polynomial model. In particular, it is known that lower stratospheric ozone in IPF v6 has a strong bias, and this influences the quality of aerosol retrievals, as ozone and aerosols are retrieved simultaneously. The AERGOM aerosol extinction was reported to show a negative bias above 25 km altitude for wavelengths larger than 700 nm (Robert et al., 2016).

In this paper, we aim at addressing these issues by using averaged GOMOS transmissions to create a multi-wavelength aerosol extinction profile dataset (climate data record). The proposed new algorithm for aerosol retrievals is based on the removal of extinctions due to the Rayleigh scattering and absorption by ozone and other trace gases from monthly and zonally averaged GOMOS transmission spectra. We aim at creating a reliable GOMOS dataset of stratospheric aerosols, which can be included into climate data records.

The paper is organized as follows. Section 2 describes the inversion algorithm. The retrieval results and various intercomparisons are presented in Section 3.  Section 4 is dedicated to characterization of the entire retrieved GOMOS aerosol dataset as a climate data record. The summary (Section 5) concludes the paper.

## 2      The inversion algorithm

### 2.1      Preparation for retrievals: creating averaged transmittances dataset

The monthly averaged transmittances $T(\lambda, z)$ as function of wavelength $\lambda$ and tangent altitude $z$ are used as a starting point for the retrievals. The self-calibrating property of the occultation method suggests averaging of independent observations for the sake of augmenting their statistical significance. This technique allows for the detection of very weak absorbers (e.g.,  OClO (Fussen et al., 2006) or Na (Fussen et al., 2010)),  as it increases the signal-to-noise ratio. Furthermore, the level of the stochastic residual scintillations is greatly reduced.

For computing averaged transmittances, we used GOMOS nighttime measurements (i.e., with a solar zenith angle at the ray perigee point larger than 107°). The refractive effects -  refractive dilution and perturbations due to scintillations  - are estimated and removed from the transmittance spectra, for each occultation (Dalaudier et al., 2001; Sofieva et al., 2009;

Kyrölä et al., 2010). The transmittances with removed refractive effects (EXT product) are used in our computations. In order to ensure sufficient number of occultations, 10° latitude zones are selected for averaging. The methodology for computing the GOMOS averaged transmittances has been developed in the dedicated ESA project ALGOM (https://earth.esa.int/eogateway/activities/algom). It includes interpolation of transmittances at a fixed altitude grid, outlier filtering, and data averaging. Because of the noise specifics, GOMOS data have outliers, which affect 2-4% of dark-limb occultations (Bertaux et al., 2010; https://earth.esa.int/eogateway/documents/20142/37627/GOMOS-Level-2-processor-version-GOMOS-6.01-Readme.pdf). The outlier filtering is performed using an algorithm based on the well-known Jackknife method ( e.g., Miller, 1974). The threshold for outliers has been chosen as 3 standard deviations from the median.

For each tangent altitude and each wavelength, the average transmittance is computed as the weighted median transmittance. This estimate is insensitive to outliers, and it takes into account the signal-to-noise ratio of different measurements, which can differ considerably for GOMOS due to different stellar sources. The transmittances are weighted with respect to the inverse of their estimated measurements errors. A classical definition of the weighted median was used (Edgeworth, 1888). Technically, it was computed as a central element of a sorted set, in which values are replicated according to their relative weights. A potential drawback of using a median instead of a mean is that it may potentially exclude isolated events such as pyrocumulonimbus events or the early stages of a volcanic eruption (note that local signals are also diluted by spatio-temporal averaging).

The examples of averaged transmittances for the latitude bin 0-10°N are shown in Figure 1, for conditions of a low loading level (January 2003, left panel of Figure 1)  and increased aerosols after Soufrière Hills and Rabaul volcanic eruptions (January 2007, right panel). It is clearly seen that the transmittances are lower (optical depth is higher) for the volcanic aerosol conditions (for example, compare transmittances at 20 km, which are highlighted by thick red lines).

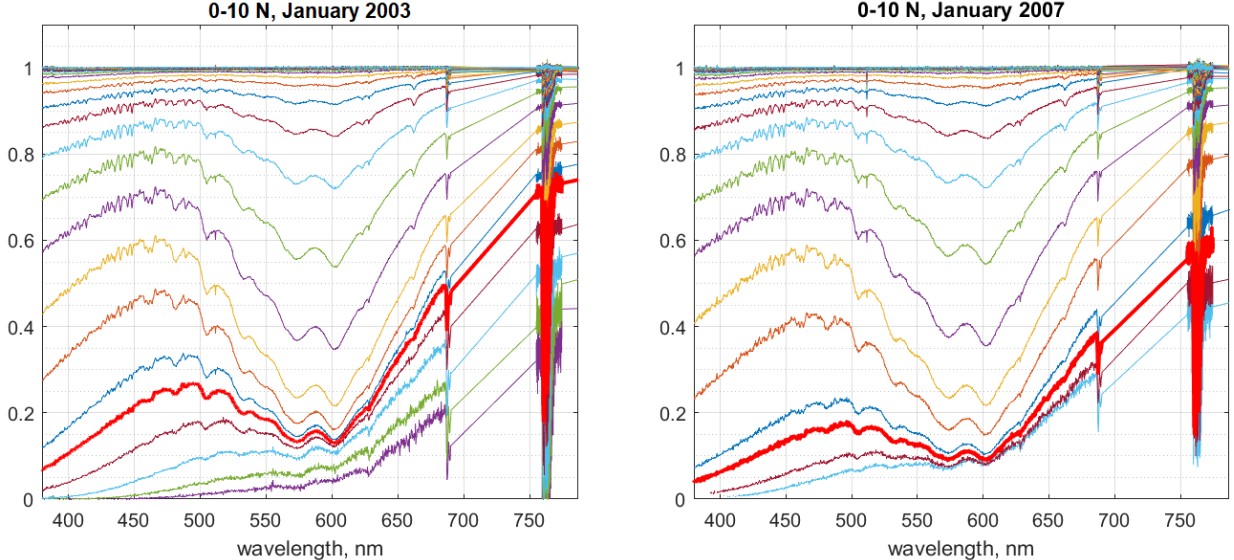

**Figure 1. Examples of GOMOS averaged transmittance spectra in the equatorial region 0-10°N, in January 2003 (background aerosols) and in January 2007 (after volcanic eruptions), from 100 km down to the lowest altitude. Red thick lines highlight transmittances at 20 km. For visibility, the transmittance spectra are plotted with the vertical step of 3 km. The lowest altitude is 8 km on the left panel and 14 km on the right panel.**

## 2.2    Aerosol retrieval algorithm

The idea of aerosol retrievals from averaged transmittances with removed refractive effects is very simple. If the extinction due to Rayleigh scattering and due to absorption by ozone, $NO_2$ and $NO_3$ can be removed from the GOMOS extinction spectra, the remaining part is due to aerosol extinction.

The GOMOS spectral inversion problem:

$$T = \exp\left(-\left(\tau_{O_3} + \tau_{NO_2} + \tau_{NO_3} + \tau_{air} + \tau_{aero}\right)\right) + \epsilon, \tag{1}$$

where $\tau_{O_3}, \tau_{NO_2}, \tau_{NO_3}, \tau_{air}, \tau_{aero}$ are contributions to the optical depth due to ozone, $NO_2$, $NO_3$, Rayleigh scattering and aerosols, respectively, and $\epsilon$ is noise, is weakly non-linear. It can be linearized by simply taking logarithm. Provided the measurements noise is small, Gaussian approximation of noise distribution for the linearized problem still holds. The GOMOS spectral inversion, its linearization and associated distribution of noise are discussed in detail in (Kyrölä et al., 1993; Tamminen and Kyrölä, 2001; Tamminen, 2004).

We used data from spectrometer A at wavelengths 380-672 nm and spectrometer B1 at wavelengths 755-759 nm. The $NO_2$ and $NO_3$ optical depths are computed using retrieved GOMOS $NO_2$ and $NO_3$ profiles, which are averaged in the corresponding month and latitude bin (Alternatively, $NO_2$ and $NO_3$ can be directly retrieved from the averaged transmittances. Since the $NO_2$ and $NO_3$ retrievals are not the subject for this study and since the standard GOMOS DOAS-

type retrieval provides good-quality $NO_2$ and $NO_3$, we use them for the aerosol retrievals). The Rayleigh extinction is computed using ECMWF air density data provided in the GOMOS files, which are averaged in the same way. As in ALGOM2S retrievals (Sofieva et al., 2017), ozone optical depth is computed using a DOAS-type retrieval with the triplet in the Chappuis band (reference wavelengths: 523 – 527 nm, 673–677 nm and absorbing wavelengths: 598–602 nm). In the processing, we filtered out unreliable averaged transmittance values (with estimated uncertainties exceeding 100 %).

The details of the retrieval of horizontal aerosol optical depth are illustrated in Figure 2 for the altitude of 20 km near the Equator (0°-10°S). The measured total optical depth, $\tau(\lambda, z) = -\log(T)$, is shown by the black line. The estimated optical depths due to ozone ($\tau_{ozone}$), $NO_2$($\tau_{NO_2}$), $NO_3$($\tau_{NO_3}$) and Rayleigh extinction ($\tau_{air}$) are shown by colored lines in the upper panel of Figure 2.

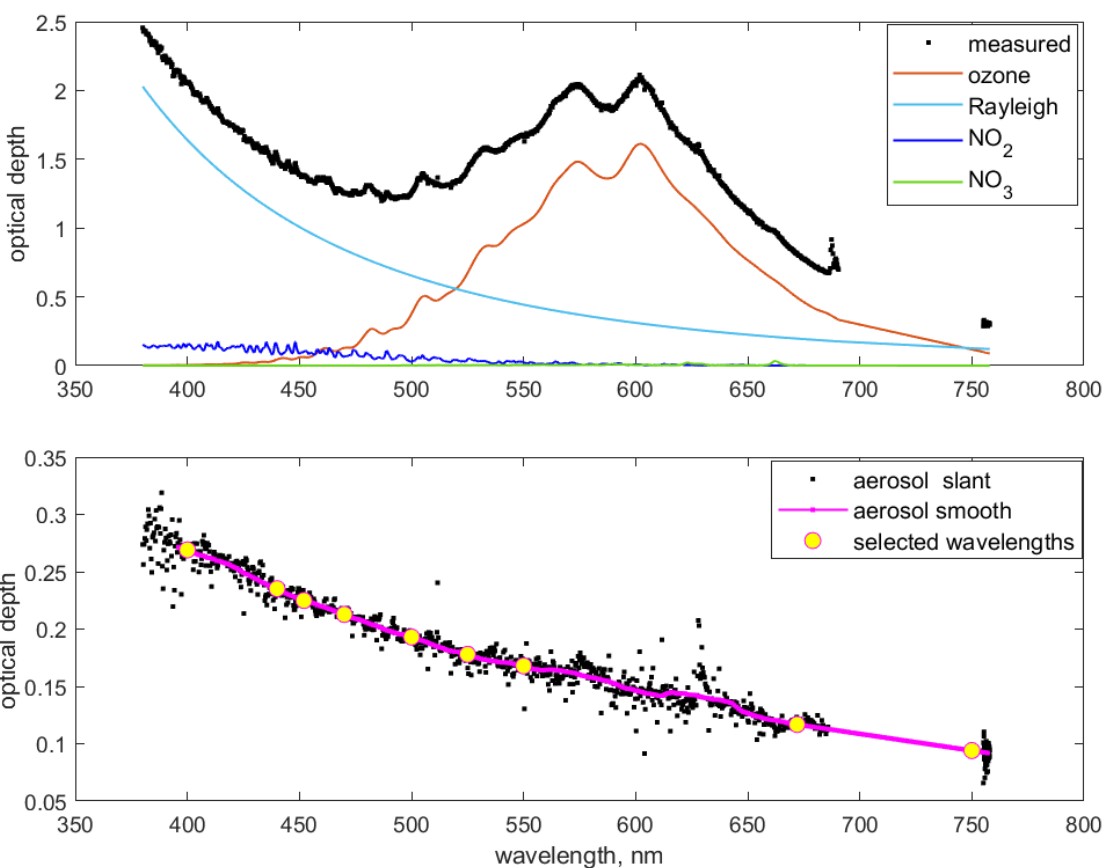

**Figure 2. Illustration of the GOMOS aerosol retrieval algorithm. Top: measured optical depth spectrum (black) and contributions due to ozone (red), Rayleigh scattering (cyan), $NO_2$ (blue) and $NO_3$(green). Bottom: original (black) and smoothed (magenta) residuals.**

The resulting residuals

$$R(\lambda, z) = \tau(\lambda, z) - \tau_{ozone}(\lambda, z) - \tau_{air}(\lambda, z) - \tau_{NO_2}(\lambda, z) - \tau_{NO_3}(\lambda, z) \qquad (1)$$

are due to aerosols. The residual spectra are smoothed and sampled at 9 wavelengths (400, 440, 452, 470, 500, 525, 550, 672 and 750 nm), resulting in the horizontal aerosol optical depth spectra $\tau_{aero}(\lambda, z)$ at each tangent altitude. The wavelengths are selected to be as used by other instruments/retrievals.

In the vertical inversion, the profiles of local aerosol extinction coefficients are reconstructed from the horizontal aerosol

optical depth profiles $\tau_{aero}(\lambda, z)$, for each wavelength. The GOMOS vertical inversion is linear, and it is performed in the same way as in the GOMOS processor (Kyrölä et al., 2010). The target-resolution Tikhonov-type regularization is applied in the vertical inversion (Sofieva et al., 2004; Kyrölä et al., 2010). The resulting actual altitude resolution of the aerosol extinction profiles $\beta(\lambda, z)$ is 3 km. The uncertainties of retrieved aerosol extinction profiles are estimated via error propagation of transmittance uncertainties. Typical uncertainties of the retrieved aerosol profiles are shown in Figure 4.

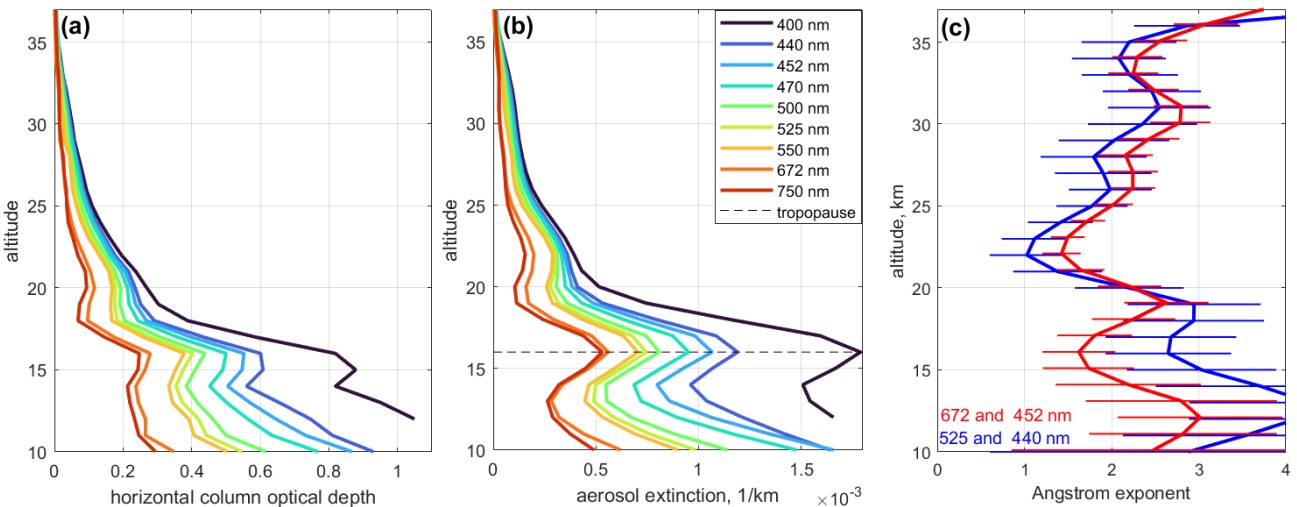


**Figure 3. The retrieved horizontal column optical depth (panel a) and aerosol extinction (panel b) for September 2002, 10°-20° S (wavelengths are indicated in the legend). Panel (c): the Ångström exponent profiles estimated using different combinations of wavelengths. The mean tropopause height is indicated by black dashed line on panel (b).**

Examples of retrieved horizontal column optical depth and aerosol extinction profiles are shown in Figure 3. The retrieved aerosol extinction is larger for shorter wavelengths, as it would be expected (e.g., Ångström, 1929; Ramachandran and Jayaraman, 2003). The profiles of the Ångström exponent

$$\alpha = - \frac{log(\frac{\beta(\lambda)}{\beta(\lambda_0)})}{log(\frac{\lambda}{\lambda_0})} \qquad (2)$$

for different wavelength combinations are shown in Figure 3c. The uncertainties of Ångström exponent are evaluated from

uncertainties of retrieved aerosol profiles via error propagation. The values of the Ångström exponent are in the expected range from 0-0.5 (large particles) to 4 (Rayleigh scattering limit). Above ~22 km, α grows with altitude becoming closer to the Rayleigh scattering limit, as expected. Below 20 km, the Ångström exponent estimates have large uncertainties, especially when using short wavelengths (example of 525 and 440 nm in Figure 3c). The Ångström exponent estimated using 672 and 452 nm are in the range 1.5- 3 below 20 km. The spectral dependence of retrieved FMI-GOMOSaero aerosol

extinction profiles is evaluated and discussed in more detail below.

## 3 Retrieval results and intercomparisons

The multi-wavelength FMI-GOMOSaero aerosol retrievals can be directly compared with data from other instruments.

In addition to other GOMOS datasets (IPF v6 and AERGOM), we compare our retrievals with aerosol data from SAGE II (Stratospheric Aerosol and Gas Experiment II) on the Earth Radiation Budget Satellite (ERBS), OSIRIS (Optical

Spectrograph and InfraRed Imaging System) on Odin and SCIAMACHY (SCanning Imaging Spectrometer for Atmospheric CHartographY) on Environmental Satellite (Envisat). The aerosol datasets used for intercomparisons are described in Table 1.

**Table 1. Aerosol datasets used for intercomparisons.**

| Instrument | Measurements/data | Retrieved aerosol parameter(s) | References |
|---|---|---|---|
| SAGE-II v.7 1984-2005 | Solar occultation, transmittances UV-VIS-IR | Aerosol extinction at 386, 452, 525 and 1020 nm | (Thomason et al., 2008) |
| GOMOS AERGOM v.4 2002-2011 | Stellar occultation, transmittances in UV VIS-NIR | Aerosol extinction at 355, 440, 470, 550 and 750 nm, aerosol size distribution | (Vanhellemont et al., 2016; Bingen et al., 2017) |
| OSIRIS v.7 (2001-present) | Limb scattering, Radiances in UV-VIS | Aerosol extinctions at 750 nm, the mode radius | (Rieger et al., 2014) |

| SCIAMACHY UBR v.3.0 (2002-2012) | Limb scattering, sun-normalized radiance at 750 nm, effective surface albedo from collocated nadir measurements | Aerosol extinction at 750 nm | The precursor retrieval version V1.4 is published in (Rieger et al., 2018). The differences to V1.4 are described in (Sofieva et al., 2023) |
| --- | --- | --- | --- |
| SCIAMACHY UBR PSD v2.0 retrievals (2002-2012) | Limb scattering, sun-normalized radiances in UV-VIS-IR, effective surface albedo from collocated nadir measurements | Particle size distribution | (Pohl et al., 2023) |


For the comparisons, the individual aerosol profiles from SAGE II, OSIRIS, SCIAMACHY and AERGOM are averaged to create monthly zonal mean data. The uncertainties of monthly mean data are estimated as the standard error of the mean.

The comparisons are focused on a potential use of the new dataset in the merged CREST dataset (Sofieva et al., 2023). Therefore, a special attention is on assessments of validity range, quality of data at 750 nm (they are from the IR spectrometer data, which are usually noisier and affected more by a combined effect of pixel response non-uniformity), and ability to reproduce geophysical variability of stratospheric aerosols.

SAGE II aerosol data are used as a reference in several studies (e.g., Rieger et al., 2015; Kovilakam et al., 2020), and SAGE II provides aerosol extinction profiles at several wavelengths. Therefore, the first example of intercomparison is with SAGE II, for latitude zone 10°-20°S in September 2002. Top panels of Figure 4 show FMI-GOMOSaero, SAGE II and AERGOM aerosol extinction profiles at several wavelengths. FMI-GOMOSaero and SAGE II aerosol extinction profiles at 525 nm agree within ±30% above 20 km (See also Figure S1 in the Supplement). Below 20 km GOMOS aerosol extinction is larger, but it is similar to SAGE II extinction with omitted cloud filtering (dashed lines in Figure 4 center). AERGOM aerosol extinction profiles also have enhancements near the tropical tropopause, while in the troposphere the AERGOM aerosol extinction coefficients become smaller at shorter wavelengths.

When using averaged transmittances, cirrus clouds cases cannot be excluded, which might result in overestimation of the aerosol extinction below 20 km (see also the discussion below).

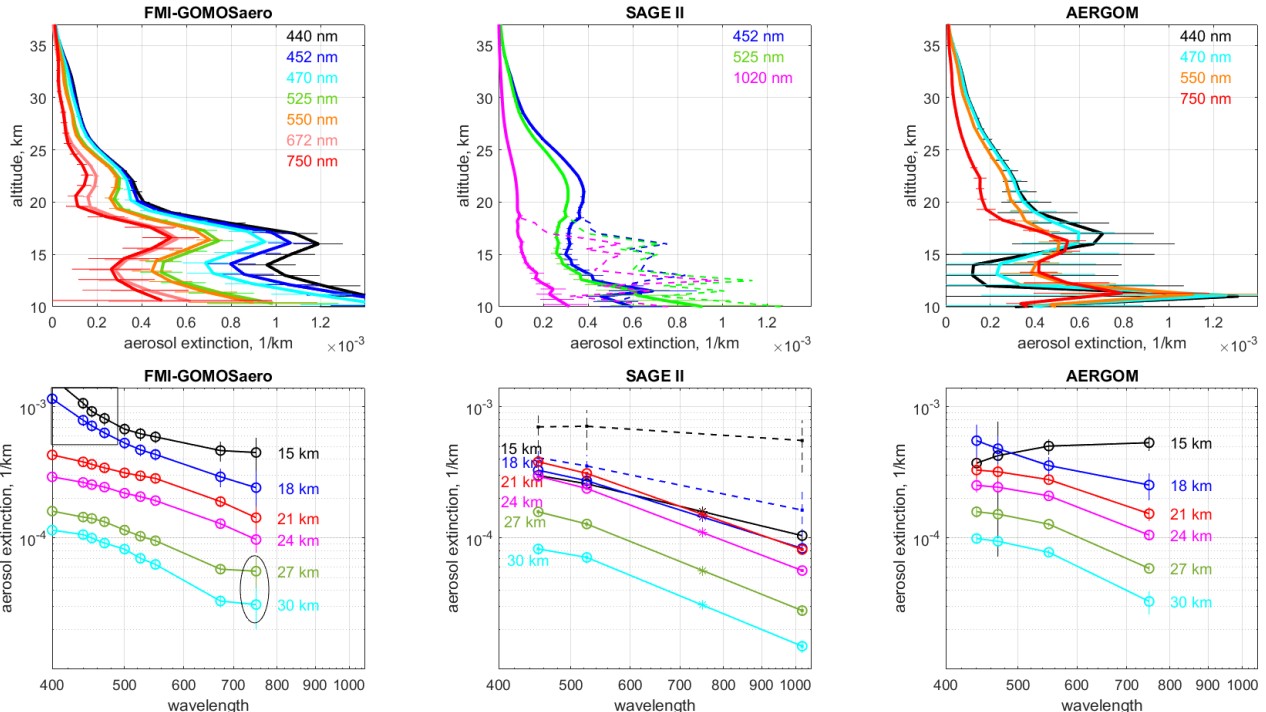

**Figure 4 Top: monthly mean aerosol profiles at several wavelengths, September 2002, 10°-20° S for FMI-GOMOSaero (left), SAGE II (center) and AERGOM v4. The wavelengths are specified in the legend. For SAGE II, profiles with (solid lines) and without cloud filtering (dashed lines) are shown. Bottom: aerosol extinction spectra at selected altitudes specified on the panels. For visual aid, colored stars on bottom middle panel indicate 750 nm aerosol extinction computed using SAGE II extinctions at 525 and 1020 nm. Oval and rectangle in the bottom left panel indicate problematic regions, which are discussed in the text.**

Bottom panels of Figure 4 show aerosol extinction spectral at several altitudes. For SAGE II, both extinctions with (solid lines) and without (dashed lines) cloud filtering are shown. For visual aid, we added values at 750 nm for SAGE II, which are computed using Ångström-exponent transformation of 525 nm and 1020 nm data, which are indicated by stars. (Note that due to curvature of aerosol extinction spectra such conversion may introduce a bias, so these 750 nm data should not be considered as measurements). We observe that the shape of aerosol extinction spectra in the lower stratosphere (21 and 24 km) is similar for all three considered datasets.

At upper altitudes, FMI-GOMOSaero spectra have a flattering at 750 nm (black oval in Figure 4). Although uncertainty of 750 nm data is large, it seems to be a systematic feature at upper altitudes. Above 30-32 km, the aerosol extinction at 750 nm is larger than that at 672 nm, and this holds for all NIR GOMOS channels (see also Figure S2 in the Supplement). Probably, this is related to compatibility of GOMOS UV-VIS and NIR spectrometers. AERGOM does not provide aerosol extinction at 672 nm, so this feature is somewhat obscured in its data, but it is revealed in Ångström exponent estimates using 750 nm channel (see below).

At lower altitudes, GOMOS data are affected by cirrus clouds, as mentioned above. Although FMI-GOMOSaero extinction spectra in the wavelength range ~525 – 750 nm have smaller slope at 15 and 18 km than at 21 and 24 km (similarly to SAGE II), the short wavelength region has, vise versa, a larger slope. At 18 km (near the tropopause), the slopes of aerosol extinction spectra are similar in FMI-GOMOSaero and AERGOM This suggest that another phenomenon can contribute to observed aerosol enhancements - imperfect refractive dilution correction due to inaccurate ECMWF data (note that very old

ECMWF forecast data have been used in the GOMOS processor). Although some improvement is observed when ERA-5 data are used, it does not completely remove the enhancements in the retrieved aerosol extinction near the tropical tropopause. In addition, the GOMOS dilution is estimated using a first-order approximation (Dalaudier et al., 2001), which might be not accurate in the vicinity of the tropical tropopause. This might be also related to imperfect estimate of Rayleigh scattering. In the troposphere (15 km), AERGOM reports a different shape of aerosol extinction, with smaller values at

shorter wavelengths. Illustrations similar to Figure 4 but for other latitude zones and time periods can be found in the Supplement.

Figure 5 shows global distribution of the average Ångström exponent in years 2003-2004 computed using SAGE II, FMI-GOMOSaero and AERGOM data. The mean tropopause, which was evaluated using ERA-5 reanalysis data, is also shown in Figure 5. Due to curvature of aerosol extinction spectra, the Ångström exponent depends on wavelengths used for its

computation. Therefore, one cannot expect numerical agreement between Ångström exponents evaluated using SAGE 525 nm and 1020 nm with those obtained from GOMOS data (it is expected to be smaller for 750&525 nm combination in case of background aerosols), therefore Figure 5 serves only for comparison of morphology of Ångström exponent. For SAGE II, Ångström exponent increases with altitude in the stratosphere, with typical values ~2 in the middle stratosphere. For FMI-GOMOSaero, similar behaviour is observed, for both combinations 672&470 nm and 672&470 nm until altitudes ~30-32

km. Above 30-32 km, we observe a decrease of Ångström exponent in the tropics and at mid-latitudes, which is strong when data at 750 nm are used. At these altitudes, data have very large uncertainty thus Ångström exponent estimate have also large uncertainty (crosses mark estimates with uncertainty exceeding 100%), so the data should be considered as unreliable in this altitude range. For AERGOM, a similar drop of Ångström exponent is observed at altitudes above 30-32km. This is consistent with analyses of aerosol extinction spectral discussed above and indicates on its instrumental origin. In the tropical

troposphere, a reduction of Ångström exponent is observed, but FMI-GOMOSaero values are larger, while AERGOM values are smaller than those of SAGE II. This region is affected by cirrus clouds, which are difficult to separate from aerosols due to a limited wavelength range of GOMOS measurements, and uncertainties of GOMOS estimates are large in this region for both GOMOS datasets. The AERGOM distribution of Ångström exponent exhibit enhanced layer above the tropopause, which is not observed in SAGE II and FMI-GOMOSaero distributions.

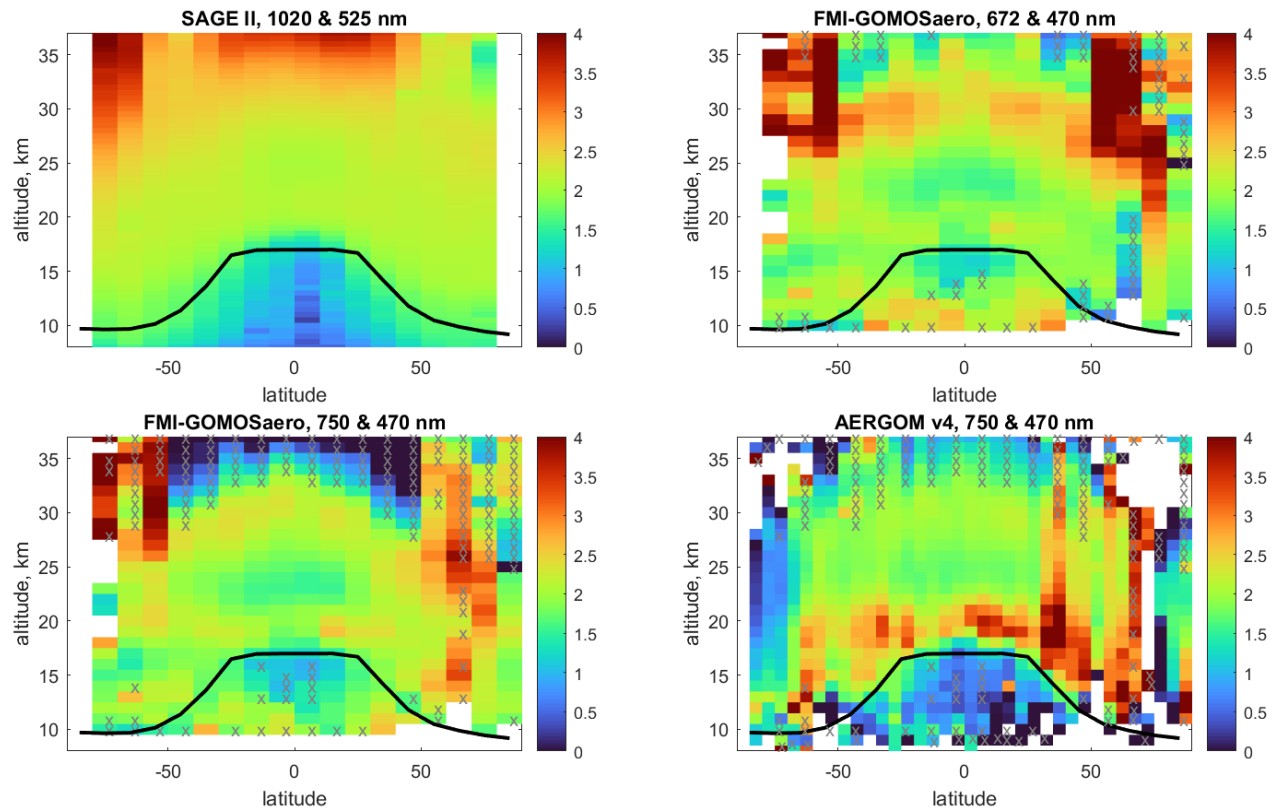


**Figure 5 Average Ångström exponent in 2003-2004 computed using SAGE II, FMI-GOMOSaero and AERGOM data. Wavelength combinations and instruments are indicated on the panels. The data with average estimated uncertainty exceeding 100% are marked with grey crosses. Black lines indicate mean tropopause. Empty boxes for AERGOM correspond to negative aerosol extinction data, for which Angström exponent has not been evaluated. SAGE II data are cloud filtered.**


In the following illustrations, we will concentrate on comparison of aerosol profiles in the stratosphere.

Examples of FMI-GOMOSaero profiles are shown in Figure 6 together with the averaged profiles from several other satellite instruments, for several latitude zones and time periods. At 525 nm, our retrievals are compared with
AERGOM (at 550 nm), SAGE II, and SCIAMACHY measurements. At 750 nm, FMI-GOMOSaero aerosol profiles are compared with AERGOM, SCIAMACHY and OSIRIS data. In both comparisons, the SCIAMACHY data from the special retrievals with the reconstruction of the particle size distribution (Malinina et al., 2018; Pohl et al., 2023) are shown with cyan lines. The blue line in the bottom panels of Figure 7 corresponds to nominal SCIAMACHY UBr v3.0 retrievals. In the bottom panels of Figure 6, SAGE II profiles, which are converted to 750 nm from 525 nm and 1020 nm using Ångström
exponent with a correction described in (Damadeo et al., 2023, Sofieva et al., 2023), are shown by black dashed lines.

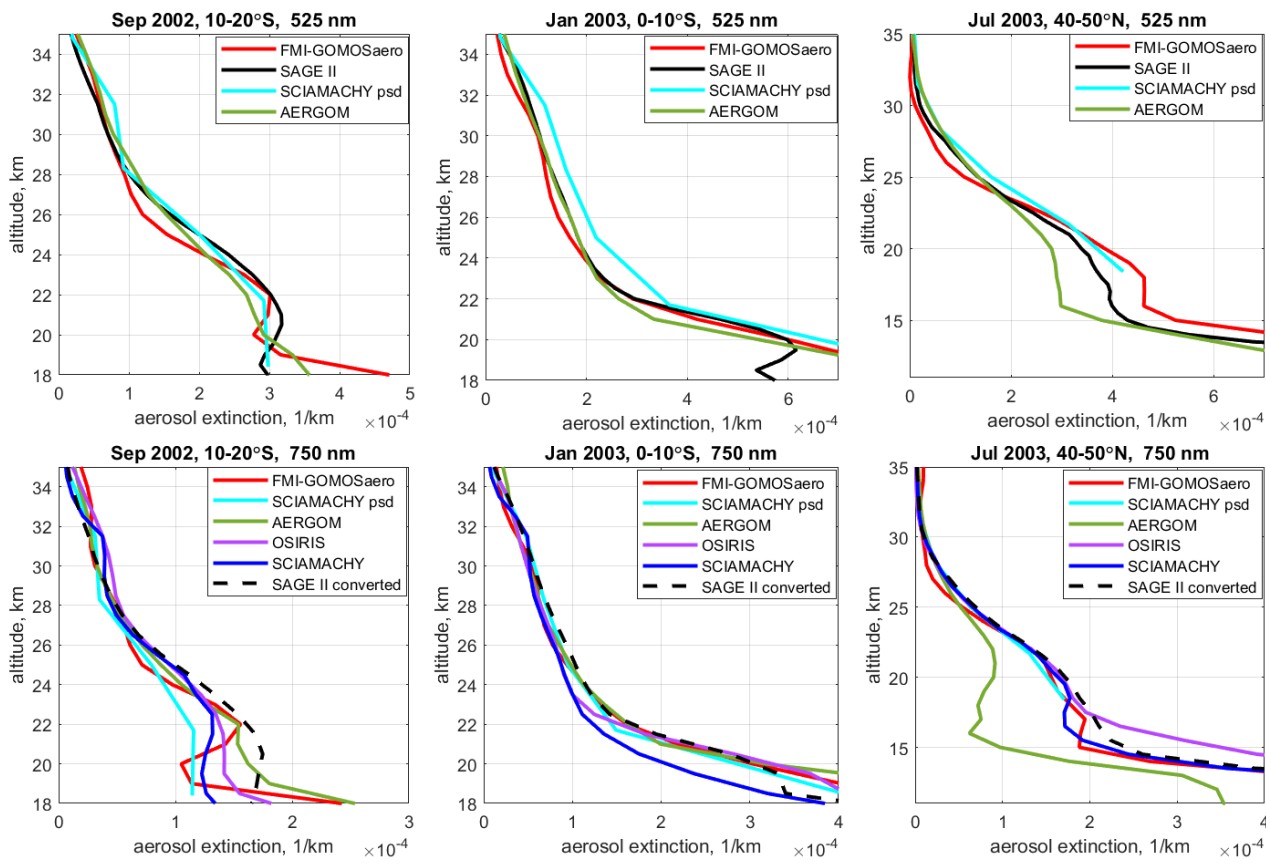

**Figure 6. Comparison of new GOMOS retrieved profiles with data from several satellite instruments. Top panels: comparisons for the wavelength 525 nm, bottom panels: comparisons for the wavelength 750 nm. The datasets are specified in the legend. The dates and latitudes are indicated in the panel titles.**

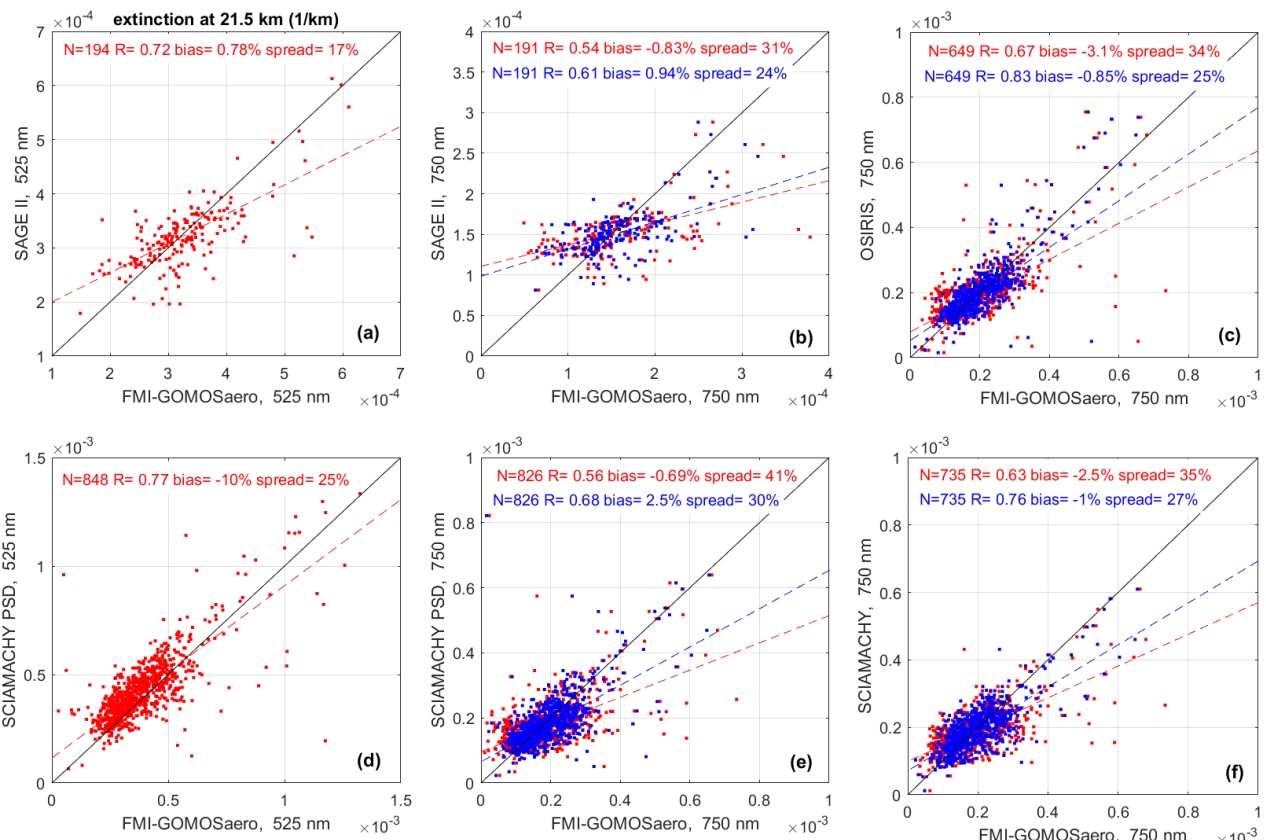

**Figure 7.** Comparison of monthly zonal mean FMI-GOMOSaero aerosol extinction coefficients at 21.5 km with analogous values from (a) SAGE II at 525 nm, (b) OSIRIS at 750 nm, (c) SAGE II converted to 750 nm using its 525 and 1020 nm data (see text for explanation), and (d) SCIAMACHY v3.0. In panels (b, c, d), the retrieved 750 nm FMI-GOMOSaero data are shown in red, while the data converted from 672 nm are shown in blue.

Figure 7 shows comparison of the monthly zonal mean FMI-GOMOSaero aerosol extinction coefficients globally at 21.5 km with analogous values from SAGE II, OSIRIS and SCIAMACHY. In Figure 7b, SAGE II data are converted to 750 nm using its 525 and 1020 nm data, as explained above. For 750 nm (Figure 7b,c,e,f), we used two collections of FMI-GOMOSaero data: retrieved at 750 nm (red) and converted from 672 nm via the Ångström exponent, which is evaluated using 672 and 500 nm (blue). In the panels, number of collocated monthly zonal mean data $N$, correlation coefficient $R$, the overall mean of relative differences $RD = \frac{FMI-GOMOSaero-instrument}{FMI-GOMOSaero+instrument} \cdot 200\%$ (bias) and their standard deviation (spread) are specified. Dashed lines are linear regression lines. As observed in Figure 7, FMI-GOMOSaero data converted from 672 nm have better correlations with the other datasets and smaller spreads with respect to them, compared to retrieved at 750 nm data. In comparisons with OSIRIS and SCIAMACHY, where both background and enhanced aerosol conditions are observed, the correlation coefficients are 0.83 with respect to OSIRIS and 0.76 with respect to SCIAMACHY. In

comparisons with SAGE II, the aerosol extinctions are smaller in the overlapping 3 years, and the correlation coefficient

between SAGE II and FMI-GOMOSaero data at 525 nm is 0.72. The overall biases at 21.5 km are less than 1 % in comparisons with SAGE II at 525nm, and a few percent in comparisons with OSIRIS and SCIAMACHY. The spread is ~20-30%.

The statistical behaviour of relative differences (RD) between FMI-GOMOSaero and other datasets, are shown for 525 nm in Figure 8 and for 750 nm in Figure 9. The differences are computed for all monthly zonal mean values available in

the datasets. The colored lines in Figure 8 and Figure 9 show the median, and horizontal bars indicate the range between 16[th] and 84[th] percentiles of the distribution. The comparisons are shown for three latitude regions: tropics 20°S-20°N (center panel), southern mid-latitudes 30°-60°S (left panel), and northern mid-latitudes 30°-60°N (right panel). In Figure 8, comparisons with SAGE II and SCIAMACHY PSD are for 525 nm. In comparison with AERGOM, both FMI-GOMOSaero and AERGOM data at 550 nm are used. The gray shading shows the region of tropical troposphere, which is affected by

clouds as discussed earlier. With respect to SAGE II and AERGOM, the biases shown in Figure 8 progress from positive values in the lowermost stratosphere to negative values above 27 km, for all latitude zones. With respect to SAGE II, FMI-GOMOSaero biases are within ±10% at 20-25 km and increase to ±20-40% at upper and lower altitudes. The biases with respect to SCIAMACHY are similar to SAGE II but negative at altitudes 18-35 km.

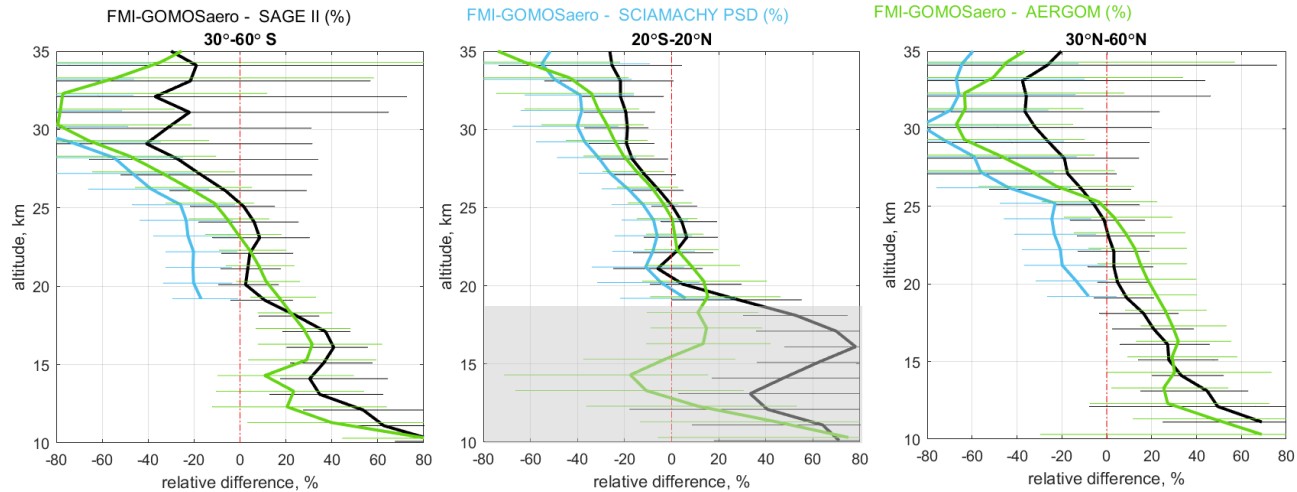


**Figure 8.  Median (solid lines) and 16th and 84th percentiles (horizontal bars) of relative differences between FMI-GOMOSaero and other datasets, for latitudes 30°-60°S (left), 20°S-20°N (center), and 30°-60°N (right).  Comparison with SAGE II and SCIAMACHY PSD are for 525 nm. In comparison with AERGOM, data at 550 nm are used. The gray shading shows the region of tropical troposphere.**


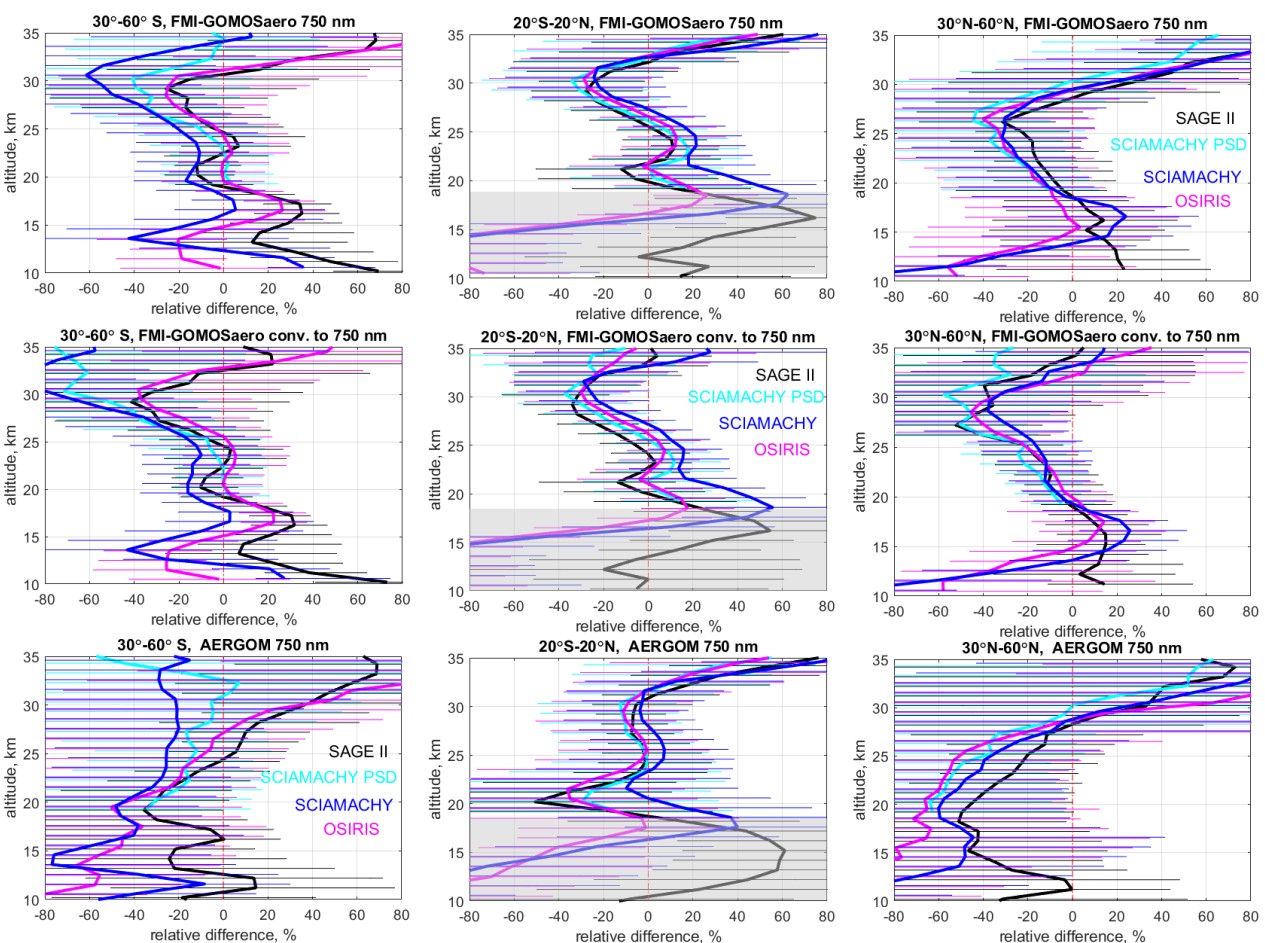

**Figure 9. As Figure 8, but comparisons are for 750 nm, and for three GOMOS datasets: FMI-GOMOSaero data retrieved at 750 nm (top panels), FMI -GOMOSaero converted from 672 nm to 750 nm (center), and AERGOM v4 at 750 nm (bottom panels). SAGE II data are converted to 750 nm (see text for the explanation).**


Figure 9 shows the analogous statistics of relative differences, but for 750 nm. We used three GOMOS aerosol datasets: FMI-GOMOSaero data retrieved at 750 nm (top panels), FMI -GOMOSaero converted from 672 nm to 750 nm (center), and AERGOM v4 at 750 nm (bottom panels). SAGE II data used in the comparison are converted to 750 nm from 525 nm and 1020 nm using Ångström exponent with a correction described in (Damadeo et al., 2023; Sofieva et al., 2023). For 750 nm,

FMI-GOMOSaero biases with respect to other datasets are mainly within 20-40% (excluding the tropical troposphere region). The best agreement and similar structure of biases are observed with repect to OSIRIS, converted SAGE II, and SCIAMACHY PSD. Comparing top and central panels of Figure 9, one can notice smaller biases and inter-percentile range for FMI-GOMOSaero data converted to 750 nm from 672 nm.

A similar analysis of relative differences, but for AERGOM v4 data is shown in bottom panels of Figure 9. The overall

structure of AERGOM biases with respect to other datasets differs from that of FMI-GOMOSaero: they are similar in the tropics, but larger at mid-latitudes. The inter-percentile range  AERGOM  relative differences to other datasets is larger, especially at mid-latitudes, which indicates  improvements in the proposed new dataset.

## 4    FMI-GOMOSaero aerosol climate data record

The entire GOMOS dataset has been processed. Subsequently, we filtered out the data points that have unrealistic values for the Ångström exponent ($\alpha>4$  below 27 km) or are potentially affected by clouds ($\alpha<-0.2$). This mild filtering is aimed at removal of strong outliers. Other data are characterized by uncertainty estimates.

By examining outliers in the processed GOMOS-FMI aerosol dataset we found that one of the reasons for strong outliers is an insufficient amount of data available for transmittance averaging.  Therefore, the data based on a small number of

averaged transmittances (less than 20) are excluded from the final GOMOS aerosol dataset.

Figure 10 shows the time series of stratospheric aerosol extinction for 672 nm at three altitude levels in the stratosphere. Main aerosol events (volcanic eruptions and wild fires) in 2002-2012 (see Table 2) are indicated in Figure 10 by black circles of the size proportional to their volcanic explosivity index VEI. Volcanic eruption signatures are clearly seen in 2007, 2009 and 2010. The enhancements are more pronounced at 17 km, as expected. The largest enhancement at 21 km

is in the tropics and it is associated with Soufrière Hills, Rabaul and Tavurvur volcanic eruptions in 2006. At 25 km, the aerosols have significantly smaller variations, one can notice gradually increasing aerosol extinction at this altitude during the GOMOS operating period.

Figure 10 also illustrates the coverage by GOMOS data. There is less data at lower altitudes; this is associated with the lowest probed altitude, which depends on stellar brightness. One can notice also a decrease of data coverage after 2009,

which is related to decreased number of GOMOS measurements due to instrumental problems (Bertaux et al., 2010).

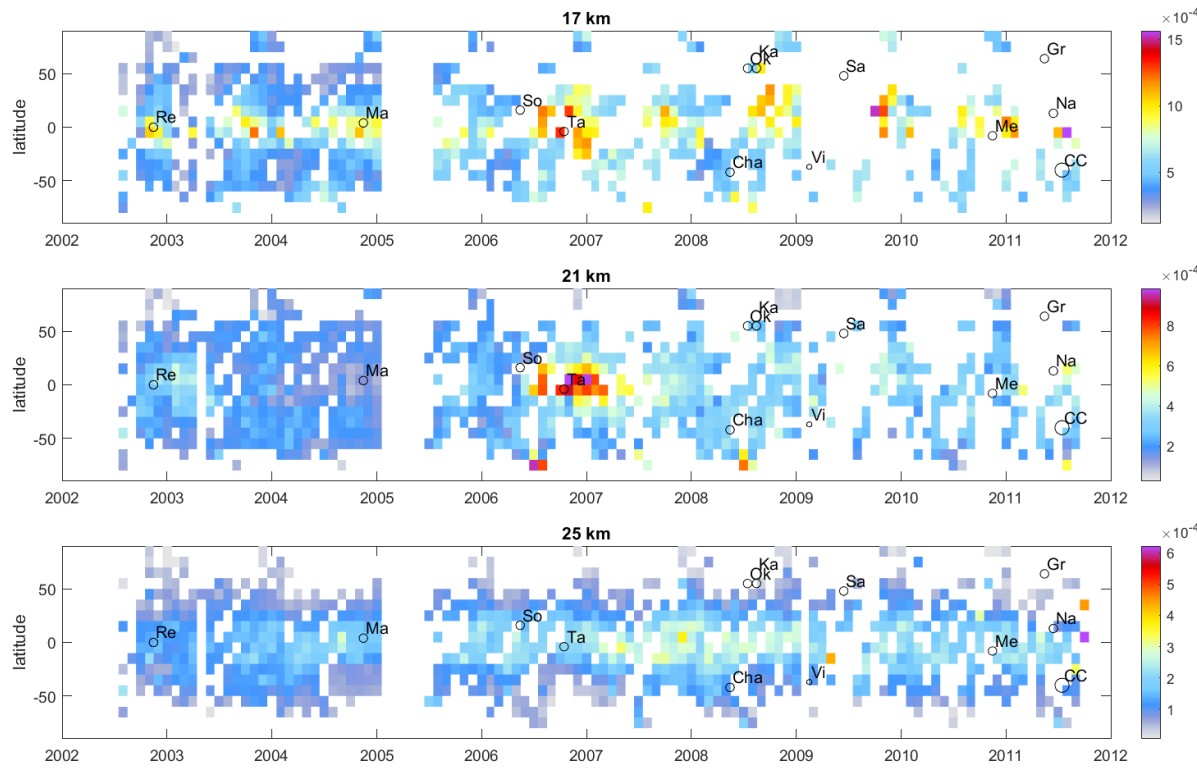

**Figure 10. Time series of aerosol extinction(1/km) at 672 nm, for 17 km (top), 20 km (center) and 25 km (bottom). Main aerosol events (volcanic eruptions and wild fires) are indicated by black circles of the size proportional to their volcanic explosivity index VEI.**

**Table 2. The list of volcanic eruptions and strong wildfires.**

| year | month | Volcano/wildfire name | Abbreviation | Latitude (deg North) | VEI | Max plume altitude |
|------|-------|-----------------------|--------------|----------------------|-----|--------------------|
| 2002 | 11 | Reventador | Re | 0 | 4 | 17 km |
| 2004 | 11 | Manam | Ma | 4 | 4 | 18-24 km |
| 2006 | 5 | Soufrière Hills | So | 16 | 4 | 17 km |
| 2006 | 10 | Rabaul/Tavurvur | Ta | -4 | 4 | 18 km |
| 2008 | 5 | Chaitén | Cha | -42 | 4 | 16.8 km |
| 2008 | 7 | Okmok | Ok | 55 | 4 | 15 km |
| 2008 | 8 | Kasatochi | Ka | 55 | 4 | 14 km |

| 2009 | 2 | Fire/Victoria | Vi | -37 | 3 | |
|------|---|---------------|-----|-----|---|---------|
| 2009 | 6 | Sarychev | Sa | 48 | 4 | 21 km |
| 2010 | 11 | Merapi | Me | -8 | 4 | 18.3 km |
| 2011 | 5 | Grimsvótn | Gr | 64 | 4 | 20 km |
| 2011 | 6 | Nabro | Na | 13 | 4 | 13.7 km |
| 2011 | 7 | Cordon Caulle | CC | -40 | 5 | 14 km |

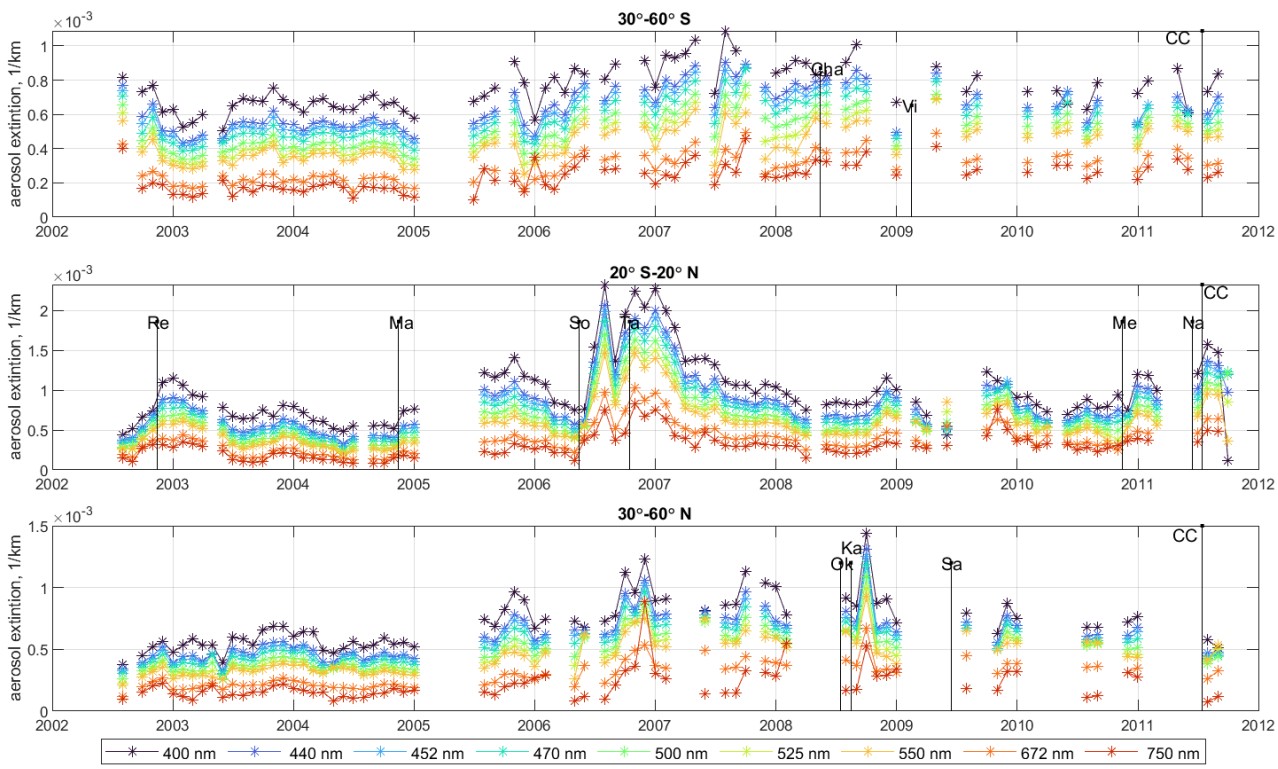

**Figure 11. Time series of GOMOS-FMI aerosol extinction (1/km) at 20 km in the latitude zones 30°-60° S (top), 20°S-20°N (center) and 30°-60° N (bottom). The wavelengths are indicated in the figure legend. The volcanos are indicated by black bars with the length proportional to volcanic explosivity index (VEI). Volcanos with VEI>=5 are shown for all latitude zones, and with VEI>3 in the corresponding latitude zones.**

Figure 11 shows the time series of aerosol extinction at 20 km in the latitude zones 30°-50° S (top), 20°S-20°N (center) and 30°-50° N (bottom), for all retrieved wavelengths. Aerosol extinction coefficient decrease with wavelength. The enhancements due to volcanic eruptions – Reventador in 2002, Soufrière Hills and Rabaul/Tavurvur in 2006, and Okmok and Kasatochi in 2008 - are seen in the data.

Figure 12 shows an analogous time series of aerosol extinction as shown in Figure 11, but using AERGOM v4 data. The AERGOM data are filtered in the same way as FMI-GOMOSaero and shown for the same latitude-time bins that are present in FMI-GOMOSaero dataset. The time series of AERGOM data without additional filtering (i.e., based on data which are reported as valid in the AERGOM files) is shown in the Supplement (Figure S5). After 2009, AERGOM aerosol extinction sometimes does not decrease with wavelength. Furthermore, AERGOM often reports negative aerosol extinction at 440 nm

and 750 nm, especially after 2009 (however, the retrieval uncertainties characterize well the AERGOM data, as discussed earlier). The enhancements related to volcanic eruptions are visible in AERGOM, but they are less pronounced compared to FMI-GOMOSaero dataset.

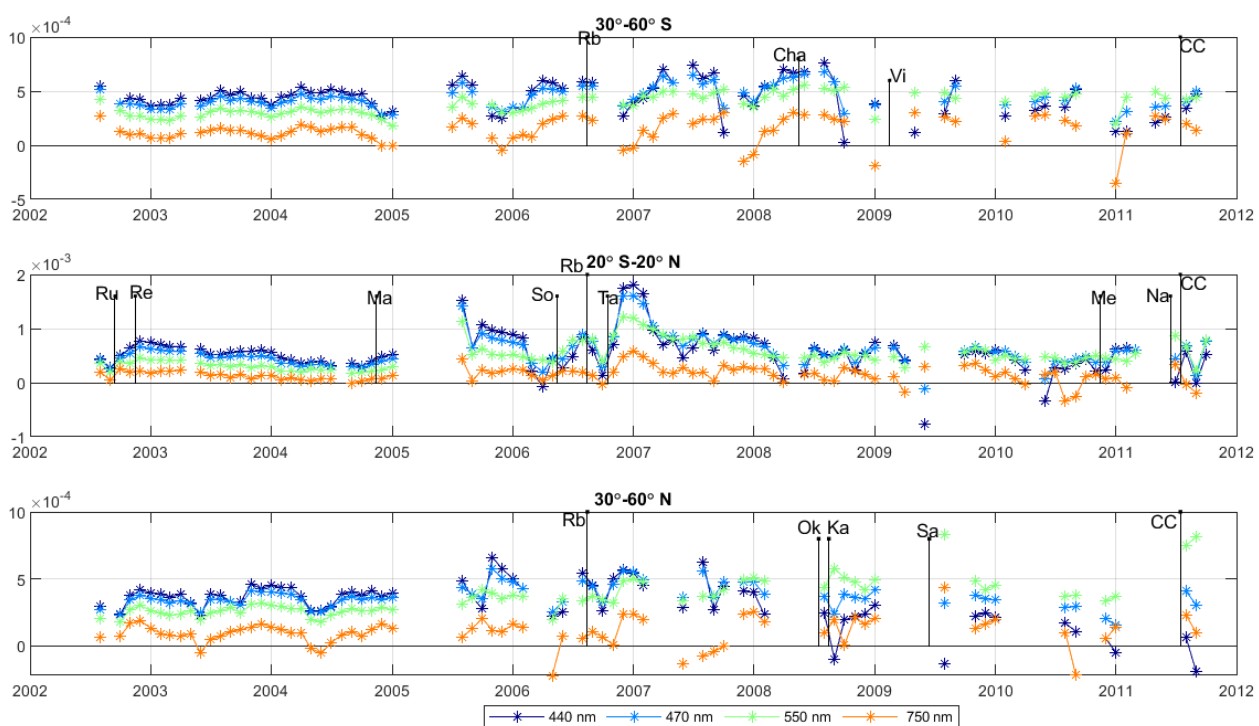

**Figure 12. As Figure 11, but for AERGOM v.4 aerosol extinction. AERGOM data are shown only for the same latitude bins and**
**times that are present in FMI-GOMOSaero data.**

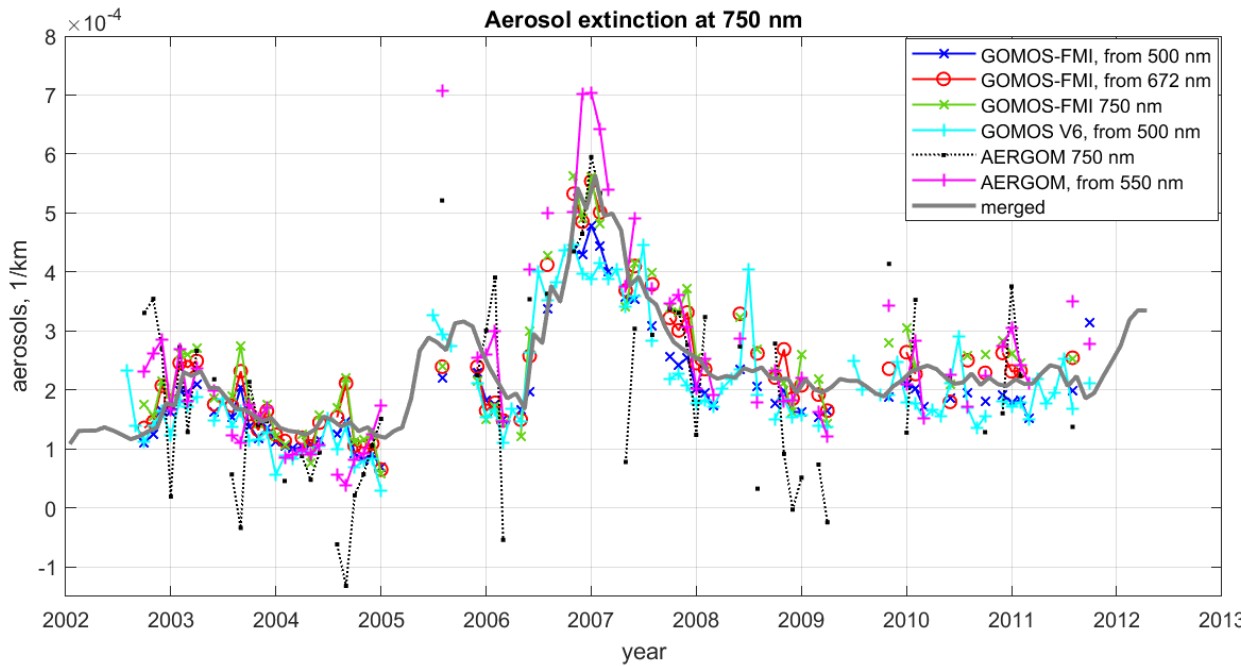

**Figure 13. GOMOS aerosol extinction at 750 nm computed by different methods: GOMOS-FMI retrieved at 750 nm (green), GOMOS-FMI converted from 672 nm (red), GOMOS-FMI converted from 500 nm (blue), GOMOS IPF v6 converted from 500 nm (cyan) and AERGOM retrieved at 750 nm(black) and converted from 550 nm (magenta). Thick grey line indicates merged aerosol extinction. All time series are for the latitude bin 0-10°N and altitude 22 km. AERGOM data are shown only for the same times that are present in FMI-GOMOSaero data.**

In order to use GOMOS aerosol profile time series in the merged Climate Data of Stratospheric Aerosols, CREST (Sofieva et al, 2023), the data at 750 nm are needed. FMI-GOMOSaero retrievals provide aerosol profiles at 750 nm. However, our intercomparison studies have shown that the FMI-GOMOSaero data converted to 750 nm from 672 nm, have smaller biases and spread, and higher correlation with to other datasets, compared to retrieved aerosol extinction coefficients at 750 nm. In general, this is not surprising, as GOMOS IR spectrometer data are noisier, and they are affected by detector non-uniformity. As an additional illustration of various GOMOS datasets, we have shown in Figure 13 the 750 nm GOMOS aerosol extinction time series, computed by different methods: either retrieved at 750 nm or computed from other wavelengths. The latter method was applied using Eq.(2) with the value of Ångström exponent 2.29, a value representative for typical stratospheric aerosol particle size distributions according to (Malinina et al., 2019). These time series, which are computed using FMI-GOMOSaero aerosols, IPF v6 and AERGOM, are presented in Figure 13, together with the merged aerosol dataset, which is computed as the median of SAGE II, OSIRIS and SCIAMACHY data (illustration of individual datasets before data merging can be found in Sofieva et al., 2023). In Figure 13, AERGOM data are shown only for the same times that are present in FMI-GOMOSaero. A similar time series but with all AERGOM data reported as valid is shown in the Supplement (Figure S2). In this illustration, FMI-GOMOSaero aerosol extinction measured at 750 nm and converted from

672 nm, are rather close to each other to the merged dataset  (and they are closer to the merged dataset than other variants of data and their conversion). However, based on statistical analyses presented in the paper, we recommend FMI-GOMOSaero data transformed to 750 nm from 672nm with the aid of  the retrieved Ångström exponent for using in the merged CREST aerosol climate data record.

## 5    Summary

The aerosol retrieval algorithm from averaged transmittance spectra has been developed and applied to the GOMOS night-time dataset. It uses the standard GOMOS two-step retrieval strategy: the spectral inversion is followed by the vertical inversion. The spectral inversion relies on the removal of contributions from ozone, $NO_2$, $NO_3$ and Rayleigh scattering from the optical depth spectra, for each ray perigee altitude. The remaining slant optical depth is due to stratospheric aerosols. In the vertical inversion, the profiles of aerosol extinction coefficients at several individual wavelengths are retrieved from the profiles of  aerosol horizontal column optical depths.

The new aerosol extinction profiles (FMI-GOMOSaero dataset v.1) are provided in the altitude range 10-40 km at wavelengths 400, 440, 452, 470, 500, 525, 550, 672 and 750 nm. The data are monthly averaged in 10° latitude bands from 90°S to 90°N, for the whole GOMOS operating period from August 2002 to March 2012. Intercomparisons of the retrieved FMI-GOMOSaero aerosol profiles with aerosol profile data from other satellite instruments found the average difference within 20-40% in the lower and middle stratosphere, the standard deviation ~ 20-50%, and the correlation coefficient of time series 0.65-0.85. Aerosol enhancements during volcanic eruptions are clearly seen in FMI-GOMOSaero aerosol time series.

The FMI-GOMOSaero monthly mean aerosol extinction profiles can be also used as a-priori information for individual GOMOS retrievals. This can be the subject of future works.

## Data availability

The FMI-GOMOSaero dataset is available at https://fmi.b2share.csc.fi/records/46080a1fce064e71975ea61c9e839721.

## Acknowledgements

The work is performed in the framework of ESA project CREST. The creation of the SCIAMACHY aerosol data set at the University of Bremen was funded in parts by ESA via CREST project, by the German Research Foundation (DFG) via the Research Unit VolImpact (grant no. FOR2820), and by the University and State of Bremen. The University of Bremen team gratefully acknowledges the computing time granted by the Resource Allocation Board and provided on the supercomputer Lise and Emmy at NHR@ZIB and NHR@Göttingen as part of the NHR infrastructure. The calculations for this research were conducted with computing resources under the project hbk00098. The FMI team thanks the Academy of Finland (Centre of Excellence of Inverse Modelling and Imaging; decision 353082).

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
