# Peer review of "Multi-wavelength dataset of aerosol extinction profiles retrieved from GOMOS stellar occultation measurements"

_Atmospheric Measurement Techniques, 2023_

## Referee Comment (RC2)

The paper describes a new aerosol dataset that is derived from the stellar occultation instrument, GOMOS. The authors use an un-conventional retrieval, which relied on zonal and monthly averaging of the level 1 transmittance, to reduce the noise levels and scintillation effect. They also compare the new product to SAGE II and SCIAMACHY aerosol profiles and GOMOS's previous retrieval. The authors could have done a better job enhancing the comparison of the new version with correlative measurements and providing some estimates of the accuracy and precision of their product. The authors should have also recommended the wavelengths that are the most accurate and most suitable for scientific studies. I find the paper suitable for publication in AMT, subject to major corrections.

Major points:

Figures 3, 4, 5, 6, and 7: The authors' extensive focus on the single averaged profile "September 2002, 10°-20° S" across five different figures, particularly since the profile primarily comprises background aerosol and clouds, seems puzzling to me. To improve the paper, I recommend consolidating these five figures into one figure with multiple panels. It might be beneficial if the authors incorporate additional cases, such as diverse volcanic eruptions and latitudes, to provide a better understanding of the new product's quality.

Figures 9, 10, and 11: Similarly, the figures are repetitive and need to say more about the quality of the FMI retrievals. The plots can easily combine into one figure with three panels. There is no need to show all 9 Wavelengths as it serves no purpose. Instead, the authors should select one reference wavelength for each retrieval, maybe 750 nm, and plot it for each zone. Additionally, displaying individual instruments rather than merged records could prevent potential distortion in the comparison, as each instrument carries its distinct biases.

Specific comments:

L73: "The proposed new algorithm for aerosol retrievals is based on the removal of extinctions due to the Rayleigh scattering and absorption by ozone and other trace gases from GOMOS transmission spectra."

This is partly true. The new algorithm is also based on averaging the transmittance zonally and monthly. Please revise the above statement.
The paper only presents the data set's average band steps in the summary section. It would be useful to include this information in section 2 as well, providing an explanation regarding the rationale for selecting these specific average band steps.

L92: "The outlier filtering is performed using an algorithm based on …"

Can you explain the cause of the outliers and how often it exists in the data?

L94: "For each tangent altitude and each wavelength, the average transmittance is computed as the weighted median transmittance. This estimate is insensitive to outliers, … "

Using a median instead of a mean may potentially exclude isolated events such as pyroCB or the early stages of a volcanic eruption, which are considered outliers by definition. The authors should address this concern within the text to acknowledge the inherent limitation of using a median for data analysis and its possible impact on capturing rare but significant events.

L102: "…increased aerosols after Soufrière Hills, Rabaul and Tavurvur volcanic eruptions"

I don't believe Rabaul and Tavurvur eruptions reached the stratosphere or anywhere near 20 km. Please check eruption altitudes and modify the text.

Figure 1: Can you clarify the start and end altitudes for each panel?

L103: "It is clearly seen that the transmittances are lower (optical depth is higher) for the volcanic aerosol conditions (compare transmittances at 20 km, thick red lines)."

The low transmittance is observable both above and below the red line. Please refine the sentence and corelate the observed changes in the transmittance with the volcanic eruption injection altitudes.

L114-L118: "The NO2 and NO3 optical depths are computed using retrieved GOMOS NO2 and NO3 profiles" "As in ALGOM2S retrievals (Sofieva et al., 2017), ozone optical depth is computed using a DOAS-type retrieval with the triplet in the Chappuis band"

It would be helpful to clarify the advantage of directly retrieving ozone while resorting to averaged $NO_2$ and $NO_3$ official retrievals, instead of either retrieving or averaging all species?

L119: "In the processing, we filtered out unreliable averaged transmittance …"

Why does the filtering process not involve individual unreliable transmittance instead of the average?

Figure 2: The selected wavelength symbol needs to be clarified. Can you choose different color and symbol?

L148: "One can notice the expected spectral dependence of aerosol extinction that is larger for shorter wavelengths."

It would be useful to provide an explanation supported by a reliable reference why this spectral dependence is expected. The 400 and 440 nm are short wavelengths, yet they show a stark difference near the tropopause.

L148-L158: The description of the profiles displayed in Figure 3 and the subsequent discussion of the Angstrom exponent lacks clarity. It would be beneficial to explicitly identify the type of aerosol profile depicted in the figure. Considering the observations, it appears that the figure shows a background aerosol layer spanning from 20-25 km and a cloud layer near the tropopause. There seems to be a discrepancy in the Angstrom exponent values, as typically, a

background aerosol layer would exhibit an Angstrom exponent range between 2 and 2.5, indicating smaller aerosol particles, rather than the values of 1 and 1.5 observed. Conversely, a near-zero Angstrom exponent would be expected for the cloud layer, contrasting with the reported values of 1.8 and 3. Addressing these discrepancies and aligning the observed values with the anticipated norms for different aerosol layers would significantly strengthen the analysis.

The repeated assertion in both the Abstract and Summary sections regarding the realistic wavelength dependence in the retrieved aerosol extinction profiles, which is based on this figure, requires additional substantiation. The authors need to provide further evidence or rationale supporting this claim.

L161: "on Earth Radiation Budget satellite, .."

Add (ERBS). Also, to be consistent, spell out ODIN and ENVISAT.

Table 1: The title "Retrieval method/data" and the entries need rectification. I propose changing it to a more accurate label, such as "Measurement/data." I also suggest specifying the types of measurements, such as solar or stellar occultation and limb scattering, corresponding to each instrument.

The information provided for OSIRIS requires correction, and it could be revised to "Radiances in UV/VIS".

There is a need to clarify the specific version used for OSIRIS aerosol data. If the authors indeed used V6.0 and not V7.0, it's important to acknowledge that V6.0 is an older, discontinued retrieval, no longer accessible online and not extending to the present time. If this is the case, the text should be amended accordingly. Additionally, retrieval method was only included for SCIAMACHY. Either include the retrieval method for all instruments or remove it from the SCIAMACHY entry to maintain uniformity.

Regarding the use of two different SCIAMACHY products for intercomparison, it could potentially lead to confusion. It would be advantageous to either select the most accurate product or explicitly justify the use of two products, providing a clear rationale for their inclusion.

Figure 4: Improvements are needed in the figure and the subsequent discussion. To enhance clarity, I recommend creating a second panel displaying the percentage difference between the two profiles. This addition will help illustrate the agreement between the profiles, particularly in regions with low aerosol values above 20 km. The existing cloud contamination around ~16 km is noticeable and may divert attention from the primary purpose of the figure, which is to compare aerosol retrievals from both instruments. My suggestion is to display solely the stratospheric portion of the aerosol profiles, reducing the interference from clouds and eliminating the necessity to show an unfiltered SAGE profile. After making these adjustments, I recommend modifying the text to further discuss the differences between the two isntruments.

L174: "Although the spectral dependence of aerosol extinction profiles is similar above 20 km, it differs below 20 km."

I don't think the reader can reach the same conclusion by looking at Figure 5. Please add an Angstrom exponent or wavelength ratio plot for both instruments (similar to Figure 6) and modify the text accordingly. As previously suggested, it's advisable to refrain from displaying the tropospheric part of the profiles.

L207: "In general, GOMOS aerosol extinction profiles are very close to those of other instruments above 20 km."

Again, I don't see how the reader can form any opinion about the differences between the FMI retrieval and other instruments just by looking at Figure 7. The figure is dominated by clouds, which is not subject to this intercomparison. As suggested above, the authors should plot only the stratospheric part of the profile and adjust the x-axis scale accordingly. They should also include two more panels showing the percent difference for each wavelength and modify the text to comment on the differences compared to each instrument reported accuracy.

L226: "we filtered out the data points that have unrealistic values for the Ångström exponent ($a>4$ below 27 km) or are potentially affected by clouds ($a<-0.2$)."

The filtering criteria, while reasonable, heavily rely on the assumption that all wavelength retrievals are consistently accurate, which might not always hold true, especially when wavelengths are in proximity to strong absorbers. As shown in Figure 3, the cloud criterion ($a<-0.2$) failed to detect the cloud near the tropopause. Please provide further insight on how effective this filter is.

Figure 8: I don't see the significance of plotting the whole record at different wavelengths rather than different altitudes, at least not by reading the text. I suggest plotting data at different altitudes instead. The discussion of the figure seems inadequate and could be enhanced. Given the extensive ten-year aerosol record, there should be interesting and noteworthy features worth highlighting.

"Table 2. The list of volcanic eruptions and strong wildfires."

In the table, there is reference to a single fire instead of multiple fires, which might be misleading. Additionally, numerous eruptions mentioned in the table seem insignificant as they did not reach the stratosphere and consequently were not detected by GOMOS. It would be advisable for the authors to revise the table by specifying the eruption altitude instead of the Volcanic Explosivity Index (VEI). Moreover, removing any low-altitude eruptions, like Eyjafjallajokull, from the table could enhance its relevance.

L272: "In order to use GOMOS aerosol profile time series in the merged Climate Data of Stratospheric Aerosols, CREST (Sofieva et al: Climate Data Record of Stratospheric aerosols, 2023, in preparation)".

If the utilization of GOMOS data in CREST records is one of the primary objectives of this paper, it should be clearly stated at the beginning of the paper for better alignment with the

study's aims. Also, remove the "in preparation" reference, as this type of reference might not be allowed.

L276: "for typical aerosol particle size distributions …"

Change to "for typical stratospheric aerosol particle size distributions …"

L281: "We found that the best agreement with the merged aerosol extinction time series is for FMI-GOMOSaero aerosol extinction converted from 672 nm."

The paper lacks a clear justification for selecting 672 nm over 750 nm. The two wavelengths look similar in Figure 11 and are shown only at a single zone and altitude. Presenting an in-depth analysis of the accuracy, or lack thereof, for both retrieved wavelengths and recommending their use in scientific studies would significantly strengthen the discussion in section 3.

L284: In the Summary and Discussion section, the title should be adjusted to "Summary and Conclusion" to accurately reflect the content. As the section contains no discussion but rather a summarization and final insights.

L297: "In the future, the developed multi-wavelength dataset of aerosol extinction profiles can be used for retrievals of particle size distribution."

The statement appears somewhat premature, particularly considering the authors lack of trust in the retrieved long wavelength, 750 nm. It would be advisable to either provide additional evidence to substantiate this statement or reconsider its inclusion.

It seems there is no section designated for Data Availability or a link provided for accessing all the data utilized in the study. A Data availability section should be included to confirm whether this data is publicly available, along with the corresponding link or repository details.

---

## Author Comment (AC1)

Dear reviewer, Dr. Robert Damadeo

Thank you very much for your comments on our paper.  We took your comments into account in the revised version of the manuscript. Please find below our detailed replies (black font) on your comments (blue font).

**Comments:**

I find it strange that one of the wavelengths from the residual spectra that is used for aerosol is one in which the instrument does not even measure (i.e., 750 nm). I understand the motivation is to match wavelengths measured by other instruments for validation purposes, but this wavelength is now essentially an extrapolation of the smoothing that is performed on the residual spectra. Additionally, why is it that the spectra shown in Fig. 2 do not cover the entire range of the IR spectrometer (thus further reducing the reliability of the smoothing out in this spectral region)?

Yes, the main reason for using 750 nm is comparison with other datasets and a potential use in the merged aerosol dataset. GOMOS measures at a very close wavelength, 755-759 nm, therefore aerosol extinction at 750 nm is very close to that of 757 nm. This is illustrated in Figure 1 below (similar to Figure 3b in the paper), when more wavelengths are included in the vertical inversion.

The IR B1 spectrometer wavelengths are for O2 retrievals. In principle, the wavelengths 770-774 nm could be also used in the aerosol retrievals.  This wavelength region is slightly noisier, and overall GOMOS IR spectrometers are affected by a combined effect of pixel response non-uniformity and intra-pixel sensitivity. Including of additional wavelengths does not improve aerosol retrievals at 750 nm. The retrieved aerosol profiles at 750 nm, 757 and 772 nm are very similar, and all three disagree with VIS spectrometer data above  ~32 km (See Figure 1 below). We discuss this disagreement in the revised version.

[Figure]

*Figure 1. Left: The retrieved aerosol extinction for September 2002, 10°-20° S (wavelengths are indicated in the legend). Right: zoom at high altitudes.*

Why not show the comparative Angstrom exponent in Figure 5 as is done for Figure 6?

In the revised version, the figures are modified. Aerosol extinction spectra are shown in new Figure 4 (which contains the information from original Figs. 5 and 6), for FMI-GOMOSaero, SAGE II and AERGOM.  Global distributions of the Ångström exponent are shown in new Figure 5.

Pg 10, Ln 210: "The reason for positive bias near the tropical tropopause is GOMOS aerosol retrievals is not fully understood at the moment."

How is it not just clouds? If you are averaging all of the transmission profiles without filtering for clouds, then of course clouds are going to bias your aerosol retrievals near the tropopause. Perhaps you cannot easily quantify how much of the bias is from clouds versus any other potential source, but the expected presence of clouds will obviously create a bias like what is shown in Fig. 7.

We corrected this statement in the revised version.

I think the better question is why does the bias appear significantly larger than the averaged SAGE II profiles in Fig. 4?

From a statistical point of view, a strong positive bias near the tropical tropopause between FMI-GOMOSaero and non-filtered SAGE II aerosol profiles is not observed.

In the revised paper,  we  restructured the discussion of biases near the tropical tropopause.

Pg 10, Ln 213: "We tried to apply various methods for cloud filtering in averaging GOMOS transmittances– according to absolute values of extinction and ratio at different wavelengths." This is no trivial task, and it may not be possible for all but the thickest clouds.

Yes, it is difficult in the GOMOS wavelength range.

As a curiosity, I wondered what the impact of using averaged transmittances would be for your comparisons. The authors compute an average transmittance profile, then convert it to an average optical depth profile, then perform the retrieval. The comparison profiles from other instruments are averages of individual profiles. As a simple test, the following relationship is true: $MEAN(-LOG(T\_i)) > -LOG(MEAN(T\_i))$. In other words, averaging the transmittance profiles first will always result in a slight low bias to your optical depths when compared with the mean of the optical depths derived from individual profiles. I am unsure of how this propagates through the two inversion steps. I would imagine the spectral inversion is less affected by this, allowing the bias to mostly propagate into the residuals. I cannot intuit how this would propagate through the vertical inversion step. If the bias still propagates proportionally (as opposed to inversely in some fashion) into the resulting extinctions, it would mean the positive bias you see in your comparisons is actually smaller than the true comparisons because a small amount of negative bias should be introduced from averaging transmissions first.

The relation  MEAN(-LOG(T_i)) > -LOG(MEAN(T_i)) is true for mean estimates. However, we use median, for which this relation does not hold.

---

## Author Comment (AC2)

Dear reviewer

Thank you very much for your comments on our paper. We took your comments into account in the revised version of the manuscript. Please find below our detailed replies (black font) on your comments (blue font).

Major points:

Figures 3, 4, 5, 6, and 7: The authors' extensive focus on the single averaged profile "September 2002, 10°-20° S" across five different figures, particularly since the profile primarily comprises background aerosol and clouds, seems puzzling to me. To improve the paper, I recommend consolidating these five figures into one figure with multiple panels. It might be beneficial if the authors incorporate additional cases, such as diverse volcanic eruptions and latitudes, to provide a better understanding of the new product's quality.

Figures 9, 10, and 11: Similarly, the figures are repetitive and need to say more about the quality of the FMI retrievals. The plots can easily combine into one figure with three panels. There is no need to show all 9 Wavelengths as it serves no purpose. Instead, the authors should select one reference wavelength for each retrieval, maybe 750 nm, and plot it for each zone.

We made the following changes in the figures.

- Previous Figure 3 has been modified by adding a panel with percentage difference and moved to Supplement.
- New Figure 4 combines the information from previous Figures 5 and 6, and with added aerosol extinction spectra.
- New Figure 5 shows global distributions of Ångström exponent.
- New Figure 6 contains the information from the original Figure 7, and shows also other examples of the data
- New Figure 7 shows scatter plots of FMI-GOMOSaero and other datasets at 21 km, for all available data
- New Figure 8 shows statistics of relative differences of FMI-GOMOSaero to other datasets at 525 nm, for 3 broad latitude zones
- New Figure 9 is similar to new Figure 8, but it shows the comparison at 750 nm, and for 3 GOMOS aerosol datasets (a multi-panel figure)
- New Figure 10 (former Figure 8) shows time series for 3 altitude levels.

The text is modified, and new discussions are added.

Additionally, displaying individual instruments rather than merged records could prevent potential distortion in the comparison, as each instrument carries its distinct biases.

The reply on this comment is below in the "Specific comments " section.

**Specific comments**:

L73: "The proposed new algorithm for aerosol retrievals is based on the removal of extinctions due to the Rayleigh scattering and absorption by ozone and other trace gases from GOMOS transmission spectra."

This is partly true. The new algorithm is also based on averaging the transmittance zonally and monthly. Please revise the above statement.

The statement is revised.

The paper only presents the data set's average band steps in the summary section. It would be useful to include this information in section 2 as well, providing an explanation regarding the rationale for selecting these specific average band steps.

We added in Section 2: "In order to ensure sufficient number of occultations, 10° latitude zones are selected for averaging."

L92: "The outlier filtering is performed using an algorithm based on …"  Can you explain the cause of the outliers and how often it exists in the data?

The cause for outliers is noise in the data, which has sometimes random-telegraph-signal (RTS) features. For dim stars, the noise level is substantial. Although RTS is removed as accurately as possible via a dark signal calibration (Bertaux et al., 2010) for each orbit, it does not always filter all outliers. According to GOMOS readme (https://earth.esa.int/eogateway/documents/20142/37627/GOMOS-Level-2-processor-version-GOMOS-6.01-Readme.pdf), outliers affect 2-4% of dark-limb data.

In the revised version, we added this information and references.

L94: "For each tangent altitude and each wavelength, the average transmittance is computed as the weighted median transmittance. This estimate is insensitive to outliers, … " Using a median instead of a mean may potentially exclude isolated events such as pyroCB or the early stages of a volcanic eruption, which are considered outliers by definition. The authors should address this concern within the text to acknowledge the inherent limitation of using a median for data analysis and its possible impact on capturing rare but significant events.

Thank you, we added this caveat to the revised version.

L102: "…increased aerosols after Soufrière Hills, Rabaul and Tavurvur volcanic eruptions" I don't believe Rabaul and Tavurvur eruptions reached the stratosphere or anywhere near 20 km. Please check eruption altitudes and modify the text.

The maximal plume rise altitude for Rabaul and Tuvurvur is 18 km, which is similar to that of Soufrière Hills (17 km).  The text is unchanged.

Figure 1: Can you clarify the start and end altitudes for each panel?

The start altitude is 100 km, and the end corresponds to the lowest altitude. The transmittance spectra in Figure 1 are plotted every 3 km. The lowest altitude is 8 km on the left panel, and 14 km on the right panel. In the revised version, we added this information to the Figure1 caption.

L103: "It is clearly seen that the transmittances are lower (optical depth is higher) for the volcanic aerosol conditions (compare transmittances at 20 km, thick red lines)." The low transmittance is observable both above and below the red line. Please refine the sentence and corelate the observed changes in the transmittance with the volcanic eruption injection altitudes.

Of course, lower transmittances are not at a single altitude. We clarified the text in order to avoid misunderstanding.

L114-L118: "The NO2 and NO3 optical depths are computed using retrieved GOMOS NO2 and NO3 profiles" "As in ALGOM2S retrievals (Sofieva et al., 2017), ozone optical depth is computed using a DOAS-type retrieval with the triplet in the Chappuis band" It would be helpful to clarify the advantage of directly retrieving ozone while resorting to averaged NO2 and NO3 official retrievals, instead of either retrieving or averaging all species?

NO2 and NO3 can be retrieved from averaged transmittances as well, with the DOAS-type retrievals.  Since the NO2 and NO3 retrievals are not the subject for the study  and since the standard GOMOS DOAS-type retrieval provides good-quality NO2 and NO3, we use this for aerosol retrievals.  We added a corresponding note in the revised version of the paper.

L119: "In the processing, we filtered out unreliable averaged transmittance …" Why does the filtering process not involve individual unreliable transmittance instead of the average?

There are several bins, where there are a few tens of occultations. When averaging them – even if there are no outliers – it might happen that averaged transmittances are still too noisy at some wavelengths. Therefore, we excluded all pixels with estimated uncertainty larger than 100% from processing.

Figure 2: The selected wavelength symbol needs to be clarified. Can you choose different color and symbol?

We improved visibility of selected wavelengths in the revised version of Figure 2.

L148: "One can notice the expected spectral dependence of aerosol extinction that is larger for shorter wavelengths." It would be useful to provide an explanation supported by a reliable reference why this spectral dependence is expected. The 400 and 440 nm are short wavelengths, yet they show a stark difference near the tropopause.

The sentence is changed to "The retrieved aerosol extinction is larger for shorter wavelengths, as it would be expected (e.g., Ångström, 1929; Ramachandran and Jayaraman, 2003).

L148-L158: The description of the profiles displayed in Figure 3 and the subsequent discussion of the Angstrom exponent lacks clarity. It would be beneficial to explicitly identify the type of aerosol profile depicted in the figure. Considering the observations, it appears that the figure shows a background aerosol layer spanning from 20-25 km and a cloud layer near the tropopause. There seems to be a discrepancy in the Angstrom exponent values, as typically, a background aerosol layer would exhibit an Angstrom exponent range between 2 and 2.5, indicating smaller aerosol particles, rather than the values of 1 and 1.5 observed. Conversely, a near-zero Angstrom exponent would be expected for the cloud layer, contrasting with the reported values of 1.8 and 3. Addressing these discrepancies and aligning the observed values with the anticipated norms for different aerosol layers would significantly strengthen the analysis.

The repeated assertion in both the Abstract and Summary sections regarding the realistic wavelength dependence in the retrieved aerosol extinction profiles, which is based on this figure, requires additional substantiation. The authors need to provide further evidence or rationale supporting this claim.

At the end of Sect.2, which describes the aerosol retrieval algorithm, we added "The spectral dependence of retrieved FMI-GOMOSaero aerosol extinction profiles is evaluated and discussed in more detail below."

In Sect.3., we provide a more detailed evaluation of the retrieved aerosol extinction profiles and their spectral dependence.

L161: "on Earth Radiation Budget satellite, .."Add (ERBS). Also, to be consistent, spell out ODIN and ENVISAT.

Added.

Table 1: The title "Retrieval method/data" and the entries need rectification. I propose changing it to a more accurate label, such as "Measurement/data." I also suggest specifying the types of measurements, such as solar or stellar occultation and limb scattering, corresponding to each instrument.

Changed as suggested.

The information provided for OSIRIS requires correction, and it could be revised to "Radiances in UV/VIS".

Corrected

There is a need to clarify the specific version used for OSIRIS aerosol data. If the authors indeed used V6.0 and not V7.0, it's important to acknowledge that V6.0 is an older, discontinued retrieval, no longer accessible online and not extending to the present time. If this is the case, the text should be amended accordingly.

In the submitted version, OSIRIS v6 was used. In the revised version, we use OSIRIS v7 data. The information is changed correspondingly.

 Additionally, retrieval method was only included for SCIAMACHY. Either include the retrieval method for all instruments or remove it from the SCIAMACHY entry to maintain uniformity.

The retrieval method for SCIAMACHY is removed.

Regarding the use of two different SCIAMACHY products for intercomparison, it could potentially lead to confusion. It would be advantageous to either select the most accurate product or explicitly justify the use of two products, providing a clear rationale for their inclusion.

The two SCIAMACHY datasets used in this study are completely independent products resulting from very different retrievals rather than different versions of the same product. While V3.0 is a direct retrieval of the aerosol extinction coefficient using an one-wavelength approach, the PSD V2.0 contains the aerosol extinction profiles calculated from the retrieved particle size distribution parameters. The latter are retrieved using a multi-wavelength approach. The former retrieval covers a wider altitude range down to about 8 km but relies on the assumed fixed particle size distribution. This assumption is, however, not considered critical during the background aerosol loading period as was the case in September 2002. The PSD V2.0 retrieval still assumes a fixed aerosol number density and retrieve the mode radius and the standard deviation of the unimodal log-normal distribution. This retrieval is limited to the altitudes above about 18 km.

As both retrievals are independent and have their own advantages and disadvantages, we think an inclusion of both results in the comparison is advantageous.

Figure 4: Improvements are needed in the figure and the subsequent discussion. To enhance clarity, I recommend creating a second panel displaying the percentage difference between the two profiles. This addition will help illustrate the agreement between the profiles, particularly in regions with low aerosol values above 20 km. The existing cloud contamination around ~16 km is noticeable and may divert attention from the primary purpose of the figure, which is to compare aerosol retrievals from both instruments. My suggestion is to display solely the stratospheric portion of the aerosol profiles, reducing the interference from clouds and eliminating the necessity to show an unfiltered SAGE profile. After making these adjustments, I recommend modifying the text to further discuss the differences between the two instruments.

This figure is moved to the Supplement. As suggested, we added a panel with percentage difference. In this figure (Figure S1) and in the updated Figure 4, the profiles are shown in full range, since they are related to discussion of influence of cloud filtering. Subsequent figures are focused on stratospheric altitudes only.

L174: "Although the spectral dependence of aerosol extinction profiles is similar above 20 km, it differs below 20 km."

I don't think the reader can reach the same conclusion by looking at Figure 5. Please add an Angstrom exponent or wavelength ratio plot for both instruments (similar to Figure 6) and modify the text accordingly. As previously suggested, it's advisable to refrain from displaying the tropospheric part of the profiles.

We changed the figure. In new Figure 4, we show SAGE II, FMI-GOMOSaero and AERGOM aerosol extinction profiles and aerosol extinction spectra. The text is modified accordingly.

L207: "In general, GOMOS aerosol extinction profiles are very close to those of other instruments above 20 km."

Again, I don't see how the reader can form any opinion about the differences between the FMI retrieval and other instruments just by looking at Figure 7. The figure is dominated by clouds, which is not subject to this intercomparison. As suggested above, the authors should plot only the stratospheric part of the profile and adjust the x-axis scale accordingly. They should also include two more panels showing the percent difference for each wavelength and modify the text to comment on the differences compared to each instrument reported accuracy.

As suggested, we show the profiles in the stratosphere in the updated version of Figure 7. We also added profiles from other latitude zones and time periods.

In addition, we added new figures that show percentage difference between datasets, for all available data, as well as scatter plots (the list of all figure changes is in the beginning of our letter).

L226: "we filtered out the data points that have unrealistic values for the Ångström exponent(a>4 below 27 km) or are potentially affected by clouds (a<-0.2)."

The filtering criteria, while reasonable, heavily rely on the assumption that all wavelength retrievals are consistently accurate, which might not always hold true, especially when wavelengths are in proximity to strong absorbers. As shown in Figure 3, the cloud criterion (a<-0.2) failed to detect the cloud near the tropopause. Please provide further insight on how effective this filter is.

This mild filtering is aimed at removal of strong outliers. Other data are characterized by uncertainty estimates. We added this note to the revised version.

Figure 8: I don't see the significance of plotting the whole record at different wavelengths rather than different altitudes, at least not by reading the text. I suggest plotting data at different altitudes instead. The discussion of the figure seems inadequate and could be enhanced. Given the extensive ten-year aerosol record, there should be interesting and noteworthy features worth highlighting.

As suggested, the new figure shows the time series at 3 altitude levels. The discussion is enhanced.

"Table 2. The list of volcanic eruptions and strong wildfires."

In the table, there is reference to a single fire instead of multiple fires, which might be misleading. Additionally, numerous eruptions mentioned in the table seem insignificant as they did not reach the stratosphere and consequently were not detected by GOMOS. It would be advisable for the authors to revise the table by specifying the eruption altitude instead of the Volcanic Explosivity Index (VEI). Moreover, removing any low-altitude eruptions, like Eyjafjallajokull, from the table could enhance its relevance.

We added the column specifying the maximum plume altitude to Table 2. We excluded from the table and from figures the low-altitude eruptions.

L272: "In order to use GOMOS aerosol profile time series in the merged Climate Data of Stratospheric Aerosols, CREST (Sofieva et al: Climate Data Record of Stratospheric aerosols, 2023, in preparation)".

If the utilization of GOMOS data in CREST records is one of the primary objectives of this paper, it should be clearly stated at the beginning of the paper for better alignment with the study's aims. Also, remove the "in preparation" reference, as this type of reference might not be allowed.

We aimed at creating reliable GOMOS dataset, which can be included into stratospheric aerosols climate data record(s). We add this note also at the beginning of the paper.

L276: "for typical aerosol particle size distributions …" Change to "for typical stratospheric aerosol particle size distributions …"

Changed.

L281: "We found that the best agreement with the merged aerosol extinction time series is for FMI-GOMOSaero aerosol extinction converted from 672 nm."

The paper lacks a clear justification for selecting 672 nm over 750 nm. The two wavelengths look similar in Figure 11 and are shown only at a single zone and altitude. Presenting an in-depth analysis of the accuracy, or lack thereof, for both retrieved wavelengths and recommending their use in scientific studies would significantly strengthen the discussion in section 3.

In the revised version, we presented more detailed analyses (the whole dataset, analyses of biases and correlation analysis), which support this conclusion.

Related to the comment:" Additionally, displaying individual instruments rather than merged records could prevent potential distortion in the comparison, as each instrument carries its distinct biases."

The last figure in our paper is dedicated to illustration of different GOMOS datasets that are either retrieved or transformed to 750 nm. As you noted, since "each instrument carries its distinct biases", it is worth to show as a reference the merged dataset, in which all data adjustments are performed already.

In the paper dedicated to data merging (Sofieva et al., 2023), there are many illustrations of original time record from various sensors. We added this note to the revised version.

L284: In the Summary and Discussion section, the title should be adjusted to "Summary and Conclusion" to accurately reflect the content. As the section contains no discussion but rather a summarization and final insights.

The title is changed to "Summary".

L297: "In the future, the developed multi-wavelength dataset of aerosol extinction profiles can be used for retrievals of particle size distribution."

The statement appears somewhat premature, particularly considering the authors lack of trust in the retrieved long wavelength, 750 nm. It would be advisable to either provide additional evidence to substantiate this statement or reconsider its inclusion.

In the revised version, this sentence is removed.

It seems there is no section designated for Data Availability or a link provided for accessing all the data utilized in the study. A Data availability section should be included to confirm whether this data is publicly available, along with the corresponding link or repository details.

We added the Section data availability and provided the link to the data (open access) in the revised version.

---

## Referee Report (RR1)

The authors did a great job addressing most of my comments, and the manuscript is vastly improved. I recommend it for publication, subject to minor changes. See below.

L110: "and increased aerosols after Soufrière Hills, Rabaul and Tavurvur volcanic eruptions".
If the authors are confident that Rabaul and Tavurvur are two distinct eruptions, please include references for each. Otherwise, if they agree with my comment regarding Table 2, then they need to modify the text accordingly.

L181: "Envisat (Environmental Satellite)." Replace it with Environmental Satellite (Envisat).

Figure 7: The x and y-axis multiplier are challenging to follow. Please use $\times 10^{-4}$ and add it to the figure caption instead of each axis.

L279: "Figure 7 shows comparison of the monthly zonal mean FMI-GOMOSaero aerosol extinction coefficients at 21.5 km"
Could you please specify whether the comparison is conducted globally or for a specific zone?

L302: Can you comment on the bias with SCIMACHY? The bias is similar to SAGE, 20-40%, however, it is always negative. A comment about the AERGOM difference is also needed. I preferred to see the AERGOM difference between the other two instruments instead, but I am okay with Fmi-GOMOS-AERGOM plot if the authors can explain its significance to the reader.

L338: "The largest enhancement at 21 km is in the tropics and it is associated with Soufrière Hills, Rabaul and Tavurvur volcanic eruptions in 2006." Also, in L361.
See my comments regarding L110 and Table 2, and make sure the text is consistent if you agree with my comments.

L326: A short statement pointing to the improvement of the proposed new dataset over the AERGOM dataset is warranted here.

Table 2: I'm not aware of a Rabaul eruption in August 2006. However, it erupted in October 2006, reaching 18 km, according to this article https://reliefweb.int/report/papua-new-guinea/volcano-erupts-papua-new-guinea-island, reporting that the eruption took place in Mount Tavurvur, near Rabaul. which may lead to different naming of the same eruption. Please disregard my comment if you can provide a reference for both eruptions in L110 or modify the text if you agree they are the same.

---

## Author Response (AR2)

Dear Editor,

Thank you very much for the attention to our paper. We thank also reviewers for their valuable comments. Please find below our replies on remaining comments.

L110: "and increased aerosols after Soufrière Hills, Rabaul and Tavurvur volcanic eruptions".
If the authors are confident that Rabaul and Tavurvur are two distinct eruptions, please include references for each. Otherwise, if they agree with my comment regarding Table 2, then they need to modify the text accordingly.

You are right, it was the same prolonged eruption with the maximum in October 2006 (https://volcano.si.edu/faq/index.cfm?question=eruptionsbyyear&checkyear=2006, https://doi.org/10.5194/essd-12-2607-2020)

We modified Table 2, figures and text.

L181: "Envisat (Environmental Satellite)." Replace it with Environmental Satellite (Envisat).

Corrected

Figure 7: The x and y-axis multiplier are challenging to follow. Please use x10-4 and add it to the figure caption instead of each axis.

The figure is improved, the axis multipliers are clearly visible now.

L279: "Figure 7 shows comparison of the monthly zonal mean FMI-GOMOSaero aerosol extinction coefficients at 21.5 km"
Could you please specify whether the comparison is conducted globally or for a specific zone?

The comparison is made globally, we added this.

L302: Can you comment on the bias with SCIAMACHY? The bias is similar to SAGE, 20-40%, however, it is always negative. A comment about the AERGOM difference is also needed.

Corresponding comments are added.

I preferred to see the AERGOM difference between the other two instruments instead, but I am okay with Fmi-GOMOS-AERGOM plot if the authors can explain its significance to the reader.

Additional panels for AERGOM are not shown in Figure 8 (like they are shown in Figure 9) because AERGOM does not provide aerosol extinction profiles at 525 nm. Therefore, some small biases with respect to SAGE II and SCIAMACHY are expected. In the right panel of Figure 8, both AERGOM and FMI-GOMOSaero are at 550 nm.

L338: "The largest enhancement at 21 km is in the tropics and it is associated with Soufrière Hills, Rabaul and Tavurvur volcanic eruptions in 2006." Also, in L361. See my comments regarding L110 and Table 2, and make sure the text is consistent if you agree with my comments.

Corrected, as indicated above.

L326: A short statement pointing to the improvement of the proposed new dataset over the AERGOM dataset is warranted here.

A short statement is added.

Table 2: I'm not aware of a Rabaul eruption in August 2006. However, it erupted in October 2006, reaching 18 km, according to this article https://reliefweb.int/report/papua-newguinea/volcano-erupts-papua-new-guinea-island, reporting that the eruption took place in Mount Tavurvur, near Rabaul. which may lead to different naming of the same eruption. Please disregard my comment if you can provide a reference for both eruptions in L110 or modify the text if you agree they are the same."

You are right, this was a mistake. It is corrected now.

I also note that the production office requests you check your figures for readability by those with common colour blindnesses, and revise accordingly. I note many of your figures use a rainbow colour bar (for heat maps or line plots) and these are problematic for those with red-green colour blindness.

The colors in Figures 2, 4, 6, 8, 11 are changed.